# Atypical meiosis can be adaptive in outcrossed *Schizosaccharomyces pombe* due to *wtf* meiotic drivers

**María Angélica Bravo Núñez[1], Ibrahim M Sabbarini[1], Lauren E Eide[1,2†], Robert L Unckless[3], Sarah E Zanders[1,4]\***

[1]Stowers Institute for Medical Research, Kansas City, United States; [2]University of Missouri-Kansas City, Kansas City, United States; [3]Department of Molecular Biosciences, University of Kansas, Lawrence, United States; [4]Department of Molecular and Integrative Physiology, University of Kansas Medical Center, Kansas City, United States

**Abstract** Killer meiotic drivers are genetic parasites that destroy 'sibling' gametes lacking the driver allele. The fitness costs of drive can lead to selection of unlinked suppressors. This suppression could involve evolutionary tradeoffs that compromise gametogenesis and contribute to infertility. *Schizosaccharomyces pombe*, an organism containing numerous gamete (spore)-killing *wtf* drivers, offers a tractable system to test this hypothesis. Here, we demonstrate that in scenarios analogous to outcrossing, *wtf* drivers generate a fitness landscape in which atypical spores, such as aneuploids and diploids, are advantageous. In this context, *wtf* drivers can decrease the fitness costs of mutations that disrupt meiotic fidelity and, in some circumstances, can even make such mutations beneficial. Moreover, we find that *S. pombe* isolates vary greatly in their ability to make haploid spores, with some isolates generating up to 46% aneuploid or diploid spores. This work empirically demonstrates the potential for meiotic drivers to shape the evolution of gametogenesis.

**\*For correspondence:**
sez@stowers.org

**Present address:** [†]Stowers Institute for Medical Research, Kansas City, United States

## Introduction

Parasites are pervasive in biology and can impose extreme fitness costs on their hosts (*McLaughlin and Malik, 2017*; *Sorci and Garnier, 2008*). Due to these fitness effects, there can be strong selection for variants of host genes that can subvert parasites (*Kutzer and Armitage, 2016*; *McLaughlin and Malik, 2017*; *Sorci and Garnier, 2008*). However, gene variants that promote host defense may be maladapted for other facets of host physiology, leading to evolutionary tradeoffs. For example, the sickle cell trait has been selected in malaria-endemic human populations as it provides heterozygous individuals some protection against the malaria-causing parasite, *Plasmodium falciparum*. However, the advantages of this allele come with a high cost as homozygotes develop sickle cell disease (*Elguero et al., 2015*; *Serjeant, 2010*).

In addition to external parasites like *P. falciparum*, organisms are also challenged by a variety of parasitic, or 'selfish', DNA sequences within their genomes (*Burt and Trivers, 2006*). Meiotic drivers are one type of selfish DNA element found throughout eukaryotes. Meiotic drive loci exploit meiosis to increase their chances of being passed on to the next generation. Rather than being transmitted to 50% of the progeny of a heterozygote, these selfish loci use a variety of tactics to promote their own transmission into up to 100% of the gametes (*Sandler and Novitski, 1957*; *Zimmering et al., 1970*). This cheating can impose a variety of fitness costs on the host (*Zanders and Unckless, 2019*). Due to these costs, variants that suppress meiotic drive can be favored by selection (*Burt and Trivers, 2006*; *Crow, 1991*; *Hartl, 1975*). Analogous to the sickle cell trait, this could lead to

evolutionary tradeoffs where variants that are suboptimal for some aspect of gametogenesis may be selected due to their ability to mitigate the costs of meiotic drivers.

In this work, we explore the potential selective pressures meiotic drivers can impose on the evolution of gametogenesis. We use the fission yeast *S. pombe* as it is infested with multiple meiotic drive genes belonging to the *wtf* (*with Tf*) gene family (*Bravo Núñez et al., 2020*; *Eickbush et al., 2019*; *Hu et al., 2017*; *Nuckolls et al., 2017*). Different natural isolates contain between 4 and 14 predicted *wtf* drivers, almost all of which are found on chromosome 3 (*Bowen et al., 2003*; *Eickbush et al., 2019*; *Hu et al., 2017*). Each *wtf* meiotic driver encodes two proteins from two largely overlapping transcripts with distinct start sites: a poison (Wtf$^{poison}$) and an antidote (Wtf$^{antidote}$) (*Hu et al., 2017*; *Nuckolls et al., 2017*). In *wtf+/wtf-* heterozygotes, all the developing meiotic products (spores) are exposed to the Wtf$^{poison}$, but only those that inherit the *wtf+* allele express the corresponding Wtf$^{antidote}$ and neutralize the poison. This allows *wtf* drivers to gain a transmission advantage into the next generation by killing the spores that do not inherit the *wtf+* allele from heterozygous (*wtf+/wtf-*) diploids (*Hu et al., 2017*; *Nuckolls et al., 2017*).

The poison and antidote proteins of a given *wtf* meiotic driver share a considerable length of amino acid sequence (>200 residues) (*Hu et al., 2017*; *Nuckolls et al., 2017*). This shared amino acid sequence may be important for a Wtf$^{antidote}$ to neutralize a given Wtf$^{poison}$ protein. Strikingly, even a mismatch of two amino acids within the C-terminus can disrupt the ability of a Wtf$^{antidote}$ to neutralize a Wtf$^{poison}$ (*Bravo Núñez et al., 2018*). The antidote of a given *wtf* gene generally does not neutralize poisons produced by other *wtf* drivers with distinct sequences (*Bravo Núñez et al., 2018*; *Bravo Núñez et al., 2020*; *Hu et al., 2017*). As a result, in diploids heterozygous for different *wtf* driving alleles, the drivers can be described as 'competing' as they exist on separate haplotypes and produce distinct Wtf proteins. When *S. pombe* isolates outcross, multiple *wtf* drivers may be in competition during gametogenesis due to rapid evolution of the gene family (*Eickbush et al., 2019*).

Here, we find that heterozygous, competing *wtf* drivers provide a selective advantage to atypical spores that inherit more than a haploid complement of *wtf* drivers. This selective advantage is due to the preferential elimination of haploid spores, as these spores are killed by the *wtf* driver(s) they do not inherit. The selected atypical spores include aneuploids, diploids, and spores inheriting *wtf* gene duplications resulting from unequal crossovers between homologous chromosomes. We use a combination of empirical analyses and modeling to demonstrate that competing *wtf* drivers generate an environment where variants that disrupt meiotic chromosome segregation can increase fitness. Finally, we show that variants that generate high numbers of atypical meiotic products may be common in *S. pombe* populations. We were unable to determine if meiotic drivers facilitated the evolution of these variants or if the high level of atypical spores produced by *S. pombe* could have enabled the success of the *wtf* drivers. Overall, this work demonstrates the capacity of meiotic drivers to impact the evolution of gametogenesis and suggests meiotic drive could have indirectly contributed to the high frequency of atypical spores generated by *S. pombe* diploids.

## Results

### The viable spores produced by outcrossed *S. pombe* diploids are frequently aneuploid or diploid

*S. pombe* cells generally exist as haploids but will mate to form diploids when starved of nutrients. If this starvation signal is continued, these diploids will proceed into meiosis resulting in the production of spores (*Egel, 2004*; *Figure 1—figure supplement 1*). Most *S. pombe* research is conducted on isogenic strains (derived from 968 h$^{90}$, 972 h–, and 975 h+) (*Fantes and Hoffman, 2016*). We refer to the isogenic lab isolate in this work as *Sp*. Most of our knowledge about *S. pombe* meiosis thus stems from studying homozygous *Sp* diploids. Relatively little is known about the meiotic phenotypes of other isolates and of heterozygotes generated by crossing different haploid isolates (referred to as 'outcrossing' here) (*Avelar et al., 2013*; *Hu et al., 2017*; *Jeffares et al., 2015*; *Zanders et al., 2014*).

There are over 100 additional *S. pombe* isolates that have been sequenced and phenotypically characterized to some extent (*Brown et al., 2011*; *Jeffares et al., 2015*; *Tusso et al., 2019*). All known *S. pombe* isolates share an average DNA sequence identity of >99% for nonrepetitive

regions, and nearly all of these isolates contain three chromosomes. Despite minimal sequence divergence between the strains, outcrossing often yields diploids that exhibit low fertility (i.e. they produce few viable spores) (*Avelar et al., 2013*; *Gutz and Doe, 1975*; *Hu et al., 2017*; *Jeffares et al., 2015*; *Singh and Klar, 2002*; *Zanders et al., 2014*). Differences in karyotype (such as chromosomal rearrangements) between isolates and pervasive meiotic drive are the two demonstrated causes of infertility in heterozygous *S. pombe* diploids (*Avelar et al., 2013*; *Hu et al., 2017*; *Nuckolls et al., 2017*; *Zanders et al., 2014*).

We previously characterized diploids generated by outcrossing *Sp* to another isolate named *S. kambucha* (*Sk*) (*Nuckolls et al., 2017*; *Zanders et al., 2014*). Although these *Sp*/*Sk* heterozygotes make few viable spores, the majority of the surviving spores are heterozygous aneuploids or diploids, as they contain both the *Sp* copy and *Sk* copy of chromosome 3 (*Zanders et al., 2014*). We refer to this phenotype as 'disomy' and refer to spores with this trait as 'disomic' (i.e. possessing two copies of a chromosome). Disomy of chromosome 3 is the only tolerable aneuploidy within *Sp*, as spores disomic for chromosomes 1 or 2, or spores lacking any of the three chromosomes are inviable (*Niwa et al., 2006*). Additionally, diploid spores, or spores possessing an extra copy of every chromosome are viable. Furthermore, the disomic state of chromosome 3 aneuploids is unstable and the extra chromosome 3 is quickly shed as the spores grow and form colonies (*Niwa et al., 2006*).

In this study, we first set out to determine if low spore viability and disomy are common amongst the viable spores produced by other outcrossed *S. pombe* diploids. We analyzed a series of heterozygous diploids made by mating different haploid natural isolates carrying genetic markers (*ade6+* or *ade6Δ::hphMX6*) linked to centromere 3 (*Figure 1A*). We also measured fertility using the viable spore yield assay. This approach calculates the number of viable spores recovered for each cell placed onto the starvation media that induces spore formation. Ideally, a healthy diploid cell would yield four viable spores. Meiotic mutants can have low viable spore yields due to spore death, but factors like experimental technique, the number of mitotic divisions a diploid completes on the starvation media, and sporulation efficiency also affect viable spore yields. Because of this, viable spore yield values are best considered to have arbitrary units and should be compared to a control experiment carried out by the same researcher.

Similar to the previously characterized *Sp*/*Sk* heterozygotes, our genetic analyses revealed that each of the novel heterozygous diploids we tested had low viable spore yields compared to homozygous diploid controls (*Figure 1B*, compare diploids 2–7 to diploids 8–12). These reduced yields are likely due to increased spore death in the heterozygotes (*Hu et al., 2017*; *Jeffares et al., 2015*; *Singh and Klar, 2002*; *Zanders et al., 2014*).

In addition to having low viable spore yields, the heterozygous diploids frequently produced spores that grew into small colonies with irregular shapes, a hallmark of aneuploidy (*Figure 1C*, *Figure 1—figure supplement 2*; *Niwa et al., 2006*). Consistent with this, 34–78% of the viable spores produced by heterozygotes were determined to be disomic for chromosome 3 as they inherited both centromere 3-linked markers (*ade6+* and *ade6Δ::hphMX6*; *Figure 1B*, diploids 2–7). The frequency of disomic spores was generally higher in heterozygotes than homozygotes. However, there was considerable variation amongst the homozygotes with spore disomy frequencies ranging from 5–46% (*Figure 1B*, diploids 8–12). In additional experiments testing 14 more homozygous natural isolates, we observed a similar large range of disomy frequencies (*Figure 1—figure supplement 3*).

## Multiple sets of competing meiotic drivers can select for disomic spores by killing haploids

We next wanted to determine why so many of the surviving spores produced by outcrossed *S. pombe* diploids were heterozygous disomes (aneuploids or diploids) for chromosome 3. We previously proposed a model in which distinct *wtf* meiotic drivers, which are nearly all on chromosome 3, were killing haploid spores (*López Hernández and Zanders, 2018*; *Zanders et al., 2014*). We hypothesized that in the presence of diverged meiotic drivers on opposite haplotypes, haploid spores will inherit only one set of drivers and be killed by the drivers they do not inherit. However, heterozygous disomic spores are more likely to inherit all the competing *wtf* drive alleles. These disomic spores should thus survive, as they will contain every Wtf$^{antidote}$ necessary to counteract the Wtf poisons (*Figure 2A*). For example, if two *S. pombe* isolates containing diverged *wtf* meiotic drivers were to mate and undergo meiosis, the drivers from one strain would be put in direct competition against the drivers in the other strain. As only spores that inherit both drivers (from both strains)

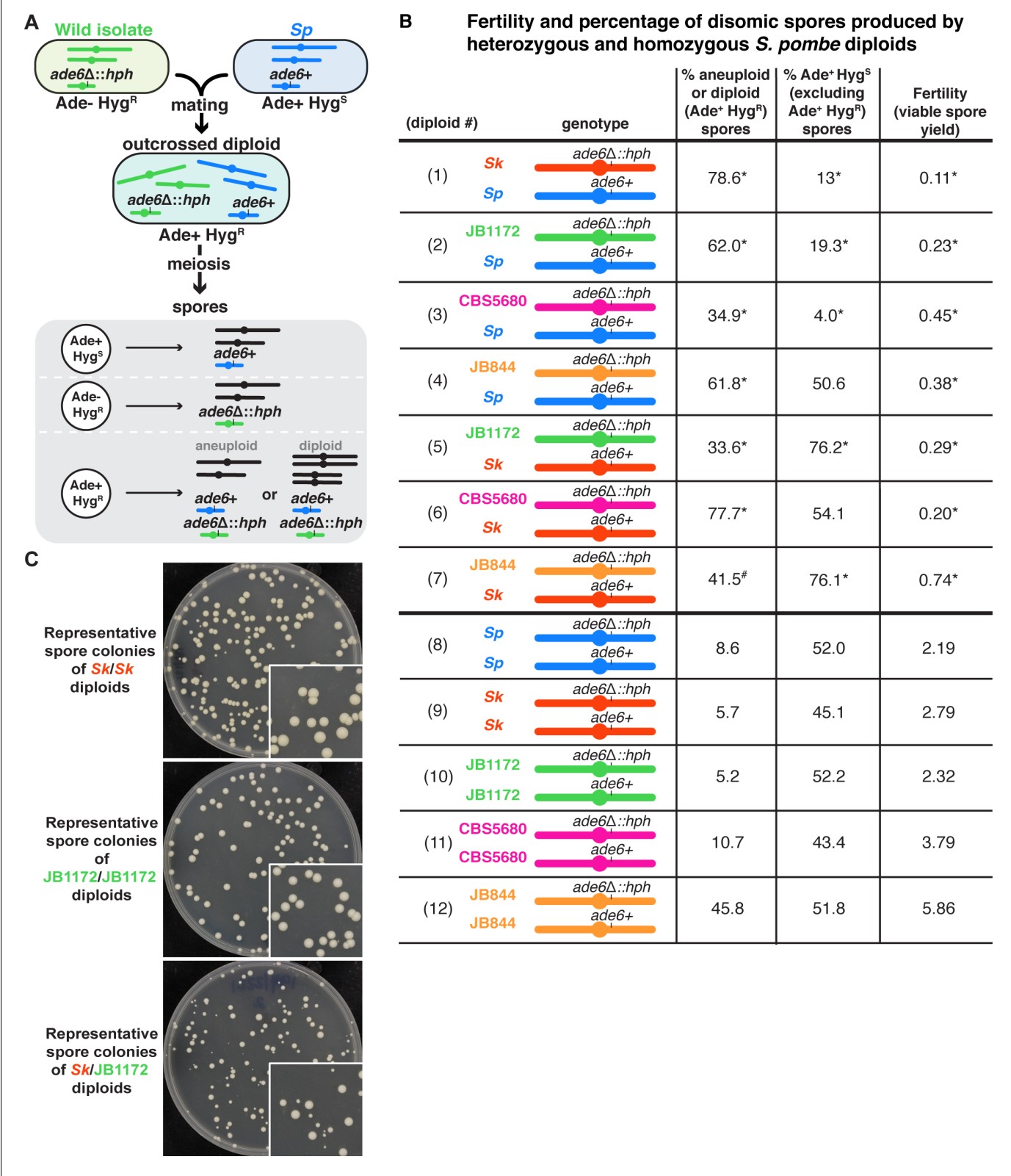

**Figure 1.** Outcrossed *S. pombe* diploids frequently produce disomic spores. (**A**) Schematic of the experimental approach. The *ade6* gene is linked to centromere 3. The karyotypes of JB1172, CBS5680, and JB844 are unknown, but outside of an inversion on *Sp* chromosome 1, *Sp* likely represents the ancestral karyotype (*Avelar et al., 2013*; *Brown et al., 2011*). (**B**) Phenotypes of heterozygous or homozygous *S. pombe* diploids. Allele transmission of chromosome 3 was assayed using co-dominant markers at *ade6* (*ade6+* and *ade6Δ::hphMX6*). The *ade6+* allele confers an Ade+ phenotype, while

*Figure 1 continued on next page*

Figure 1 continued

the $ade6\Delta$::hphMX6 provides resistance to Hygromycin B (Hyg$^R$). Heterozygous aneuploid or diploid spores are Ade+ Hyg$^R$. The fertility was measured using the viable spore yield assay to determine the number of viable spores per viable diploid. In the absence of drive, we expect 50% of the spores to be Ade+ Hyg$^S$. A significant departure from 50% indicates drive favoring the overrepresented allele. The phenotypes of each heterozygote were compared to those of homozygous diploids from both parental strain backgrounds. * indicates p-value<0.025 (G-test [Ade+ Hyg$^R$ spores] and Wilcoxon test [fertility]) for the heterozygotes relative to the homozygous diploids from both parental backgrounds. Diploid 7 was only significantly different (p-value<0.025) in the frequency of Ade+ Hyg$^R$ spores when compared to diploid 9, but not when compared to diploid 12. This is indicated with #. To detect biased allele transmission (Ade+ Hyg$^S$), diploids 2–4 were compared to diploid 8 and diploids 5–7 were compared to diploid 9. * indicates p-value<0.05 (G-test [allele transmission]). More than 200 viable spores were scored for each diploid. Raw data can be found in *Figure 1—source data 1* and *Figure 1—source data 2*. (C) Representative images of the viable spore colonies generated by homozygous *Sk* and JB1172 diploids and heterozygous *Sk*/JB1172 diploids. Images of colonies generated by other diploids are shown in *Figure 1—figure supplement 2*.

The online version of this article includes the following source data and figure supplement(s) for figure 1:

**Source data 1.** Raw data for the viable spore yield reported in *Figure 1*.
**Source data 2.** Raw data of allele transmission values reported in *Figure 1*.
**Figure supplement 1.** The *Schizosaccharomyces pombe* life cycle.
**Figure supplement 2.** Colony phenotypes of spores produced by *S. pombe* heterozygous and homozygous diploids.
**Figure supplement 3.** Homozygous *S. pombe* diploids generate variable frequencies of disomic spores.
**Figure supplement 3—source data 1.** Raw data of allele transmission values reported in *Figure 1—figure supplement 3*.

would survive the toxicity of the Wtf poisons, the resulting population would be enriched for these atypical meiotic products. Consistent with this model, *Sp/Sk* heterozygotes do not make more disomic spores per meiosis than *Sp* or *Sk* homozygotes (*Zanders et al., 2014*).

To test our model, we engineered an *Sk* diploid that is heterozygous for two unlinked sets of *wtf* drivers on chromosome 3. This diploid was generated by crossing two *Sk* strains that, apart from the different *wtf* drivers we integrated, are isogenic. We refer to this diploid as the 'double driver heterozygote' (*Figure 2B*). Importantly, all four of the drivers in this strain are functional and cannot fully suppress any of the other three drivers (*Figure 2—figure supplement 1*; *Bravo Núñez et al., 2018*; *Bravo Núñez et al., 2020*; *Nuckolls et al., 2017*). Consistent with our hypothesis, we found that 63% of the viable spores generated by the double driver heterozygote appeared to be disomic as they inherited both alleles at *ade6* (*ade6*-:FY29033 *wtf36:natMX4* and *ade6*-:*Sp wtf13:hphMX6*). This was considerably higher than the control diploid (no heterozygous *wtf* drivers) in which 4% of the viable spores displayed this phenotype (*Figure 2C*, compare diploid 13 to 14). Additionally, many of the viable spores produced by the double driver heterozygote generated small, misshapen colonies characteristic of aneuploids. Finally, the fertility of the double driver heterozygote was 20-fold lower than the control diploid (*Figure 2C*, compare diploid 13 to diploid 14). To test if our results were dependent on strain background, we made an analogous double driver heterozygote in the *Sp* strain background and observed a similar decrease in fertility and increase in disomy amongst the surviving spores (*Figure 2—figure supplement 2*). These phenotypes are consistent with the destruction of haploid spores that do not inherit every driver.

We also considered an alternative hypothesis that meiosis in the double driver heterozygotes more frequently produces disomic spores. The Wtf$^{poison}$ proteins can be detected at low levels prior to the meiotic divisions and could thus theoretically alter the fidelity of chromosome segregation (*Nuckolls et al., 2017*). However, it is important to note that every *S. pombe* meiosis likely occurs in the presence of multiple distinct Wtf$^{poison}$ proteins. This makes it difficult to imagine how heterozygosity for competing *wtf* drivers could specifically interfere with chromosome segregation. Nevertheless, we tested if expressing distinct Wtf$^{poison}$ proteins from a heterozygous conformation promoted the production of disomic spores. To do this, we used separation-of-function alleles (*Sk wtf28$^{poison}$* and *Sk wtf4$^{poison}$*) that produce the Wtf$^{poison}$ proteins, but not the corresponding Wtf$^{antidote}$ proteins. We found that *Sk wtf28$^{poison}$*/*Sk wtf4$^{poison}$* heterozygotes had very low viable spore yields, but did not exhibit a higher frequency of disomic spores than control diploids (*Figure 2—figure supplement 3*). This is consistent with our previous analyses of hemizygous (*wtf$^{poison}$*/*ade6*+) mutants in which we did not observe a high frequency of disomic spores (*Bravo Núñez et al., 2018*; *Nuckolls et al., 2017*). These results argue against a role for Wtf proteins in affecting chromosome segregation.

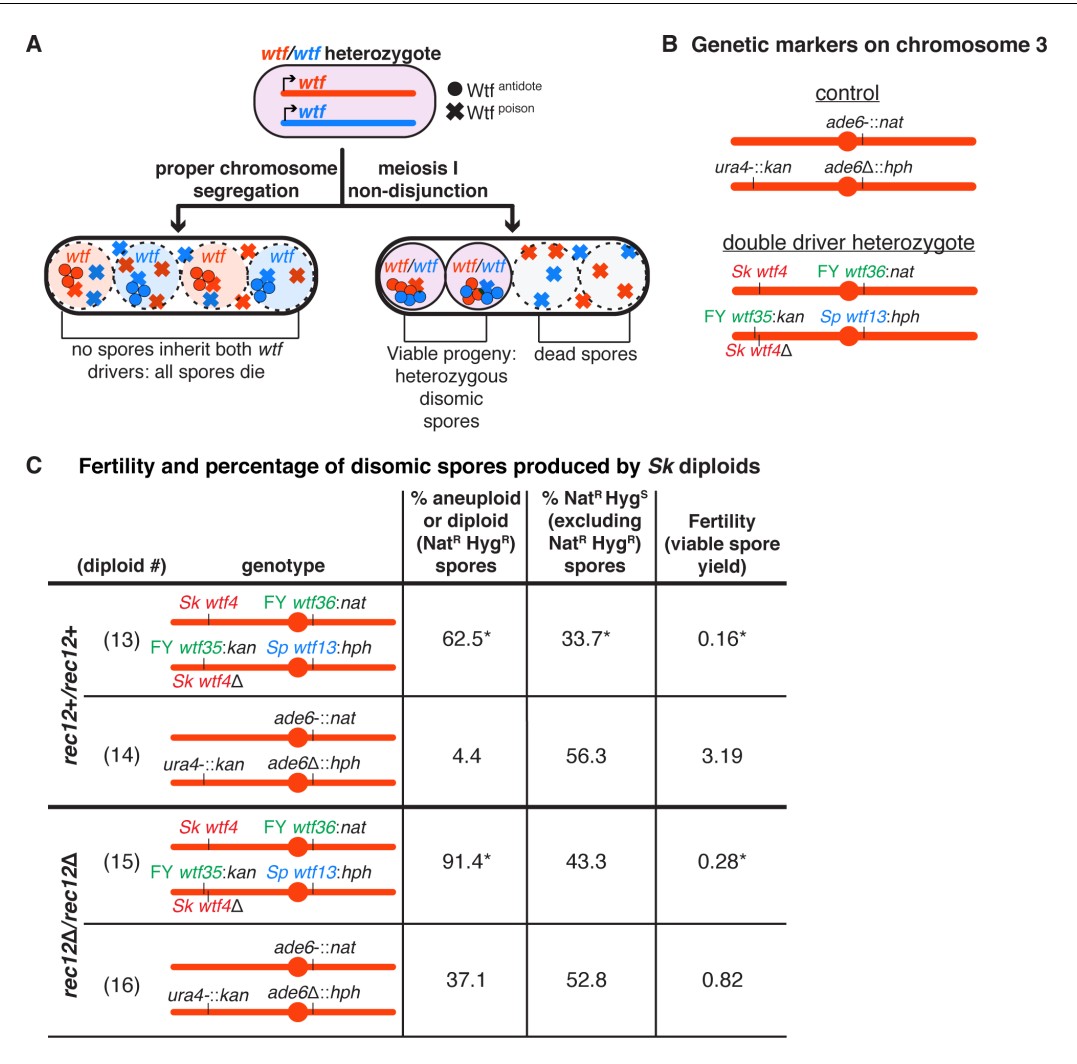

**Figure 2.** A high fraction of viable spores are disomic in *Sk* strains with *wtf* competition at two loci. (**A**) Model for a diploid heterozygous for distinct *wtf* meiotic drivers. Spores are destroyed by any *wtf* driver that they do not inherit from the diploid progenitor cell. Meiosis I chromosome missegregation is one mechanism by which spores can inherit *wtf* alleles on competing haplotypes and survive. (**B**) Schematic of the genetic markers at *ura4* and *ade6* in the control diploid and the *wtf* transgenes inserted at *ura4* and *ade6* in *Sk* chromosome 3 in the double driver heterozygote. *wtf* genes from the *Sp*, *Sk*, and FY29033 strains are depicted in blue, red, and green, respectively. The *wtf* drivers shown here drive when heterozygous and do not counteract the effect of the other drivers (see *Figure 2—figure supplement 1*). (**C**) Phenotypes of the double driver heterozygote or control diploid in *rec12*+ (top) and *rec12Δ* (bottom) strain backgrounds. We expect Nat^R Hyg^S spores to be present at 50% in the viable population. A significant departure from the expected 50% indicates drive favoring the overrepresented allele. For statistical analyses, the frequency of disomic spores, allele transmission, and fertility in the double driver heterozygotes was compared to the control diploids. Diploid 13 was compared to control diploid 14, and diploid 15 was compared to control diploid 16. * indicates p-value<0.05 (G-test [allele transmission and Nat^R Hyg^R spores] and Wilcoxon test [fertility]). The data for diploid 14 were previously published in *Bravo Núñez et al., 2020*. Raw data can be found in *Figure 2—source data 1* and *Figure 2—source data 2*. The online version of this article includes the following source data and figure supplement(s) for figure 2:

**Source data 1.** Raw data of allele transmission values reported in *Figure 2*, *Figure 2—figure supplement 1*, *Figure 2—figure supplement 2*, and *Figure 2—figure supplement 3*.

**Source data 2.** Raw data of viable spore yield assays reported in *Figure 2*, *Figure 2—figure supplement 1*, *Figure 2—figure supplement 2*, and *Figure 2—figure supplement 3*.

**Figure supplement 1.** Wtf^antidote proteins are generally specific for a Wtf^poison and do not provide cross-resistance to other Wtf^poison proteins.

**Figure supplement 2.** *Sp* diploids with *wtf* competition at two loci have a high fraction of viable spores that are disomic.

**Figure supplement 3.** *wtf* poison-only alleles do not enrich for disomes in viable spores.

**Figure supplement 4.** Competing *wtf* meiotic drivers do not affect the frequency of disomes generated per meiosis.

**Figure supplement 5.** The recombination frequency is altered in *Sk* diploids with competing *wtf* meiotic drivers.

In addition, we used the observed frequency of disomes and the viable spore yield to calculate the number of disomes produced per diploid cell placed on sporulation media. This calculation approximates the number of disomic spores generated per meiosis. We found that the number of disomic spores produced per cell placed on sporulation media did not increase between the double driver heterozygote and the control diploid (*Figure 2—figure supplement 4*, compare diploid 13 to diploid 14). These results are also inconsistent with the hypothesis that *wtf* drivers affect the frequency at which meiosis generates disomic spores.

Together, our results are consistent with our hypothesis that disomic spores generated from diploids carrying multiple sets of heterozygous *wtf* meiotic drivers are enriched in the surviving population due to the destruction of haploid spores, rather than due to Wtf proteins increasing the rate of meiotic chromosome missegregation.

## Heterozygosity at *wtf* loci contributes to the high frequency of disomic spores generated by outcrossed diploids

To further determine the contribution of competing *wtf* drivers to the high level of disomic spores observed in the viable progeny, we decided to test our model in a strain background with more extensive heterozygosity for *wtf* drivers, like those generated by outcrossing. For these experiments, we started with an *Sp/Sk* mosaic diploid strain that is heterozygous for eight known or predicted *wtf* meiotic drivers (*Bravo Núñez et al., 2020*; *Eickbush et al., 2019*). This mosaic diploid is

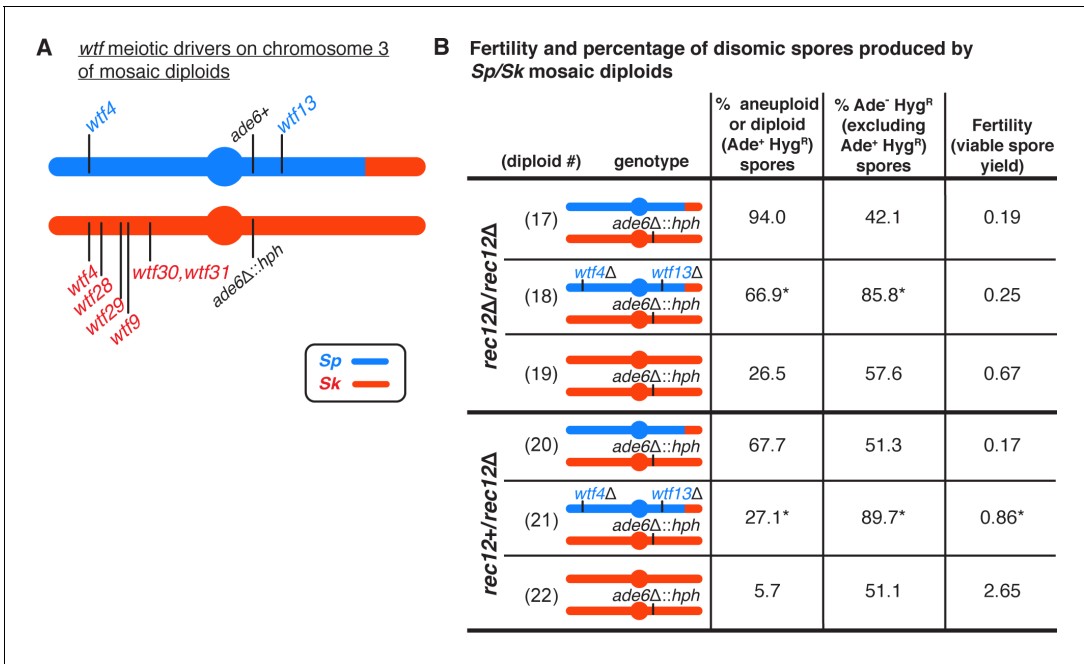

**Figure 3.** *wtf* meiotic driver competition contributes to the high disomy in spores produced by *Sp/Sk* mosaic diploids. (**A**) Schematic of the predicted *wtf* meiotic drivers found on chromosome 3 of the *Sp/Sk* mosaic diploid. *Sp*-derived DNA is depicted in blue and *Sk*-derived DNA in red. (**B**) Phenotypes of mosaic and control diploids in *rec12Δ/rec12Δ* and *rec12+/rec12Δ* backgrounds. Allele transmission of chromosome 3 was assayed using markers at *ade6* (linked to centromere 3). In the absence of drive, we expect 50% of the spores to be Ade- Hyg[R]. Any significant deviation from the expected 50% indicates drive favoring the overrepresented allele. To determine the contribution of *wtf* meiotic drivers to the frequency of disomic spores and fertility, diploid 18 was compared to diploid 17, and diploid 21 was compared to diploid 20. To determine if there was biased allele transmission, diploids 17 and 18 were compared to control diploid 19, and diploids 20 and 21 were compared to control diploid 22. More than 300 viable spores were scored for each diploid. * indicates p-value<0.05 (G-test [allele transmission and Ade+ Hyg[R] spores] and Wilcoxon test [fertility]). Raw data can be found in *Figure 3—source data 1* and *Figure 3—source data 2*.

The online version of this article includes the following source data and figure supplement(s) for figure 3:

**Source data 1.** Raw data of allele transmission values reported in *Figure 3* and *Figure 3—figure supplement 1*.
**Source data 2.** Raw data of viable spore yield reported in *Figure 3* and *Figure 3—figure supplement 1*.
**Figure supplement 1.** Single deletions of *Sp wtf* meiotic drivers partially decrease the frequency of disomic spores produced by *Sp/Sk* mosaic diploids.
**Figure supplement 2.** Observed recombination frequencies are altered by meiotic drive in *Sp/Sk* mosaic diploids.

homozygous for *Sk* chromosomes 1 and 2 but is heterozygous for most of chromosome 3 for *Sp*- and *Sk*-derived sequences (*Figure 3A*). These diploids also lack *rec12*, a gene which encodes the endonuclease that initiates meiotic recombination by generating DNA double-strand breaks (DSBs) (*Bergerat et al., 1997*; *Keeney et al., 1997*). The lack of induced recombination in these diploids results in absolute linkage within the *Sp* and *Sk* haplotypes on chromosome 3. Because of this, haploid spores will generally inherit either every *Sp* driver or every *Sk* driver. We again used markers at the centromere-linked *ade6* locus to determine the transmission of *Sp*- and *Sk*-haplotypes on chromosome 3.

Consistent with our previous observations in a similar mosaic diploid (*Nuckolls et al., 2017*), we saw that the viable spores generated by this diploid were almost exclusively (94%) heterozygous disomes for chromosome 3 (*Figure 3B*, diploid 17). Recombination promotes faithful segregation of chromosomes, so the lack of recombination in this mosaic diploid likely contributed to the high disomy we observed amongst the viable spores. Lack of recombination is, however, insufficient to explain the majority of the phenotype, as *rec12Δ Sk* homozygous diploids generate only 27% disomic spores (*Figure 3B*, diploid 19).

To test if the extremely high frequency of disomic spores generated by the *Sp/Sk* mosaic diploid was dependent on *wtf* meiotic drivers, we deleted the predicted *Sp* drivers, *Sp wtf13* and *Sp wtf4*. This eliminated *wtf* driver competition as the remaining *wtf* drivers were either on the same haplotype or homozygous. Consistent with our hypothesis, deleting both *Sp* drivers significantly decreased the frequency of chromosome 3 heterozygous disomes (from 94% to 67%; *Figure 3B*, diploids 17 and 18), although it did not significantly increase fertility. Deleting only one of the two *Sp* drivers was also sufficient to significantly decrease the frequency of disomic spores (*Figure 3— figure supplement 1*, diploids 47 and 48). However, deleting only one of the six predicted *Sk* drivers (*wtf4*) had no effect (*Figure 3—figure supplement 1*, diploid 49). This was expected as considerable *wtf* driver competition remains as *Sk* still has five intact drivers.

We also performed analogous experiments in the presence of meiotic recombination by mating the mosaic haploid strain to a *rec12+ Sk* strain. Meiotic recombination will produce chromosomes with new combinations of *Sp* and *Sk wtf* drivers. Despite these new combinations, our model predicts that heterozygous disomic spores will still have a fitness advantage as they are more likely to inherit every *wtf* driver. We observed that this *rec12+/rec12Δ* diploid had low fertility, similar to that of the *rec12Δ Sp/Sk* mosaic diploid (*Figure 3B*, compare diploid 17 to diploid 20). To assay disomy amongst the spores produced by the *rec12+/rec12Δ* mosaic diploid, we again genotyped the *ade6* locus (*Figure 3A*). We found that 68% of the viable spores generated by this mosaic appeared to be disomic for chromosome 3. Deleting both *Sp wtf4* and *Sp wtf13* in the *rec12+/rec12Δ* mosaic diploid significantly increased fertility and decreased disomy at *ade6* amongst the viable spores from 68% to 27% (*Figure 3B*, diploids 20 and 21).

We also examined single deletions of *Sp wtf4* or *Sp wtf13* in a Rec12+ mosaic diploid. Deleting *Sp wtf4* or *Sp wtf13* individually decreased disomy amongst the viable spores, but only the *Sp wtf4* deletion significantly increased fertility (*Figure 3—figure supplement 1*, diploids 50 and 51). These results demonstrate that *wtf* driver competition contributes to the extremely high frequency of disomes amongst the surviving spores and can contribute to low spore viability in these mosaic strain backgrounds. However, *wtf* competition alone was insufficient to explain the total increase in disomy relative to the *Sk* homozygotes (*Figure 3*, compare diploid 18 to diploid 19, and diploid 21 to diploid 22). Overall, our results support the model that high disomy observed in the spores generated by outcrossed *S. pombe* diploids is partially due to competing *wtf* meiotic drivers.

## Driver landscapes affect observed recombination rates on collinear haplotypes

Meiotic drivers are often associated with regions of suppressed recombination, such as chromosomal inversions (*Dobzhansky and Sturtevant, 1938*; *Dyer et al., 2007*; *Hammer et al., 1989*; *Larracuente and Presgraves, 2012*; *Pieper and Dyer, 2016*; *Stalker, 1961*; *Svedberg et al., 2018*). This state of recombination suppression is thought to be indirectly caused by the driver as linked loci also enjoy a transmission advantage. However, there is little empirical evidence about how the presence or absence of drivers can directly affect recombination landscapes. Fortuitously, our experiments allowed us to address this question by assaying how drivers can affect recombination in otherwise isogenic strains.

We first assayed recombination between the *ade6* and *ura4* loci of the double driver heterozygote and control diploid described in *Figure 2B*. The *ade6* and *ura4* loci are over 75 cM apart in the *Sk* control (*Figure 2—figure supplement 5*). In the *Sk* double driver heterozygotes, this distance decreased significantly to 44 cM (p-value=0.03, G-test). We hypothesize this is because recombination can uncouple two of the strongest drivers, *Sp wtf13* and FY29033 *wtf35*, which are found on the same haplotype.

We also analyzed the effect of drivers on recombination in the *rec12+/rec12Δ* mosaic diploids described in *Figure 3*. In the mosaic diploids with all drivers intact, we observed *ade6* and *ura4* were 43 cM apart (*Figure 3—figure supplement 2*), similar to the 62 cM previously observed in *Sp/Sk* hybrids (*Zanders et al., 2014*). However, when *Sp wtf4* and *Sp wtf13* were deleted from the mosaic strain, the observed genetic distance decreased significantly to 11 cM (*Figure 3—figure supplement 2*; p-value<0.01, G-test). We hypothesize this drop is due to preferential death of recombinants, as recombination would lead to haploids failing to inherit every *Sk* driver. Overall, our results demonstrate that meiotic drivers can directly affect recombination landscapes.

## Meiotic driver competition at a single locus selects for atypical meiotic products

The experiments above test scenarios with at least two sets of competing *wtf* drive genes. However, when more closely related isolates mate, the number of heterozygous *wtf* driver loci will be reduced. We analyzed this scenario by measuring the impact of one set of competing meiotic drivers in a diploid. To do this, we tested an *Sp* diploid heterozygous for *Sk wtf4* and *Sk wtf28* transgenes integrated at the *ade6* locus (*Figure 4A*). We compared this diploid to a control heterozygote (empty vectors at *ade6*). We found that *Sk wtf4/Sk wtf28* heterozygous diploids had decreased fertility (13% of the control diploid; *Figure 4C* and *Figure 4—figure supplement 1*) and 77% of the viable spores appeared to be disomic for chromosome 3 as they inherited both *Sk wtf4*- and *Sk wtf28*-linked drug resistance markers (G418[R] Hyg[R] spores, *Figure 4C*). These phenotypes are not specific to those drivers, the *Sp* strain background, or the *ade6* locus. We also observed similar phenotypes in an *Sk* background and with drivers at the *ura4* locus (*Figure 2—figure supplement 1* and *Figure 4—figure supplement 2*). As in our previous experiments, these phenotypes are consistent with the destruction of spores that do not inherit both *wtf* drivers leading to the enrichment of spores that inherit both drivers amongst the survivors.

Although the spores produced by diploids with one set of competing meiotic drivers (*Sk wtf4/Sk wtf28*) often inherited both *wtf* driver-linked drug resistance alleles, they generally did not exhibit a colony morphology typical of aneuploids, as we observed in our other experiments (*Figure 1C*, *Figure 1—figure supplement 2*). We, therefore, investigated the ploidy of the spores of this cross more thoroughly using colony morphology, sporulation phenotypes, phloxin B staining, and a genetic marker loss assay (see Materials and methods, *Figure 4—figure supplement 3*). We were surprised to discover that amongst the G418[R] Hyg[R] spores generated by the *Sk wtf4/Sk wtf28* heterozygote, only 15.4% appeared to be aneuploid and none appeared to be diploid. Instead, the majority (84.6%) of the G418[R] Hyg[R] spores appeared to be haploid (*Figure 4C*, diploid 24). We reasoned that an unequal interhomolog crossover event at the *ade6* locus could have led to duplication of the *wtf* driver found on the opposite haplotype (*Figure 4—figure supplement 4*). Such a crossover would allow a haploid spore to inherit both *wtf* drivers and thus be protected against the toxicity of Wtf poisons. In our previous experiments, a single crossover could not put all the drivers onto one haplotype (*Figure 2* and *Figure 3*).

Consistent with the idea that a crossover placed the competing *wtf* drivers on one haplotype in the *Sk wtf4/Sk wtf28* diploid meiosis, the events were Rec12-dependent (*Figure 4*, diploid 26). In addition, we detected a PCR product consistent with the recombination event in 19 out of the 22 haploid G418[R] Hyg[R] spore colonies tested (*Figure 4*, diploid 24, *Figure 4—figure supplement 4*, and *Figure 4—figure supplement 5*). We detected similar unequal crossover products amongst the G418[R] Hyg[R] haploid spores generated by the control diploid as well, but at much lower frequencies (*Figure 4C*, diploid 23 and *Figure 4—figure supplement 5*). Therefore, we concluded that this type of atypical meiotic product (duplication due to an unequal crossover) was enriched amongst the spores of the *Sk wtf4/Sk wtf28* heterozygote due to the death of spores that did not inherit both *wtf* drive genes.

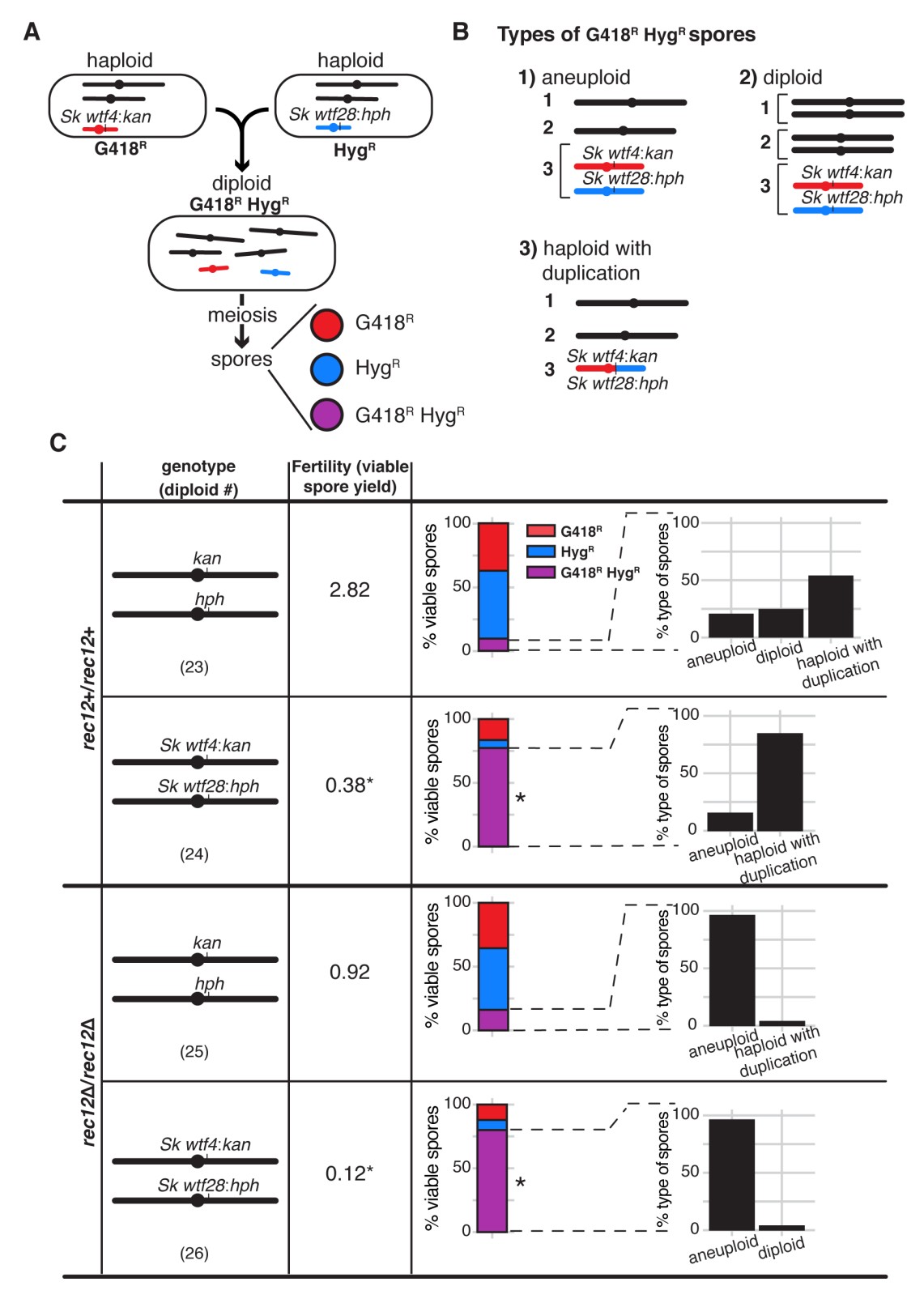

**Figure 4.** *wtf* competition at a single locus in meiosis selects for spores that are aneuploid, diploid, or contain a *wtf* duplication. (A) Schematic of *Sp* strains crossed to make an *Sk wtf4:kanMX4/Sk wtf28:hphMX6* heterozygote (diploid 24) and the possible spore phenotypes produced by that diploid. A similar control diploid (diploid 23) was created with empty vectors linked to drug resistant markers at the same locus. (B) Types of G418^R Hyg^R spores. We distinguished these classes using a series of phenotypic and molecular tests (see Materials and methods, *Figure 4—figure supplement 3*,

*Figure 4 continued on next page*

Figure 4 continued

*Figure 4—figure supplement 4*, and *Figure 4—figure supplement 5*). (C) Viable spores observed in control crosses (vector/vector) or with competing *wtf* meiotic drivers at the *ade6* locus in a *rec12*+ (top) or *rec12Δ* (bottom) background. Percentages of G418$^R$ Hyg$^R$ (aneuploid, diploid, haploid with a duplication event) spores are shown. For statistical analyses, we compared diploid 24 to control diploid 23, and diploid 26 to control diploid 25. * indicates p-value<0.05 (G-test [G418$^R$ Hyg$^R$] and Wilcoxon test [fertility]). Raw data can be found in *Figure 4—figure supplement 5*, *Figure 4—source data 1*, and *Figure 4—source data 2*. The data for diploid 23 [excluding inset analyses of G418$^R$ Hyg$^R$ spores] were previously published in *Bravo Núñez et al., 2020*.

The online version of this article includes the following source data and figure supplement(s) for figure 4:

**Source data 1.** Raw data of allele transmission values reported in *Figure 4C* and *Figure 4—figure supplement 2*.
**Source data 2.** Raw data of viable spore yield reported in *Figure 4* and *Figure 4—figure supplement 2*.
**Figure supplement 1.** Most spores are destroyed by *wtf* driver competition.
**Figure supplement 2.** Competing *wtf* meiotic drivers enrich for spores that contain both *wtf* alleles in *Sk*.
**Figure supplement 3.** Methods to distinguish between haploid, diploid, and aneuploid spore colonies.
**Figure supplement 4.** Unequal crossover event at *ade6* leads to *wtf* duplication.
**Figure supplement 5.** Evidence of unequal crossover events between transgenes inserted at *ade6*.

## Fitness costs of meiotic mutants can be mitigated or eliminated in diploids with competing *wtf* drivers

Our results demonstrate that when distinct *wtf* drivers compete on different haplotypes, such as when *S. pombe* outcrosses, the atypical spores that inherit more drivers are more fit. Disomic spores that inherit two copies of chromosome 3 most likely inherit the maximal number of *wtf* drivers, as most *wtf* drivers are found on that chromosome. Therefore, we hypothesized that the fitness costs of decreasing the fidelity of meiotic chromosome segregation might be offset by the fitness benefits of generating more disomic spores when *wtf* drivers compete (*Figure 5A and B*). Consistent with this idea, we previously observed that deleting *rec12* imposed no fitness cost on *Sp/Sk* heterozygotes compared to the *Sp/Sp* or *Sk/Sk* homozygotes (*Zanders et al., 2014*; *Figure 5C*). We wanted to know if this was specific to *Sp/Sk* heterozygous diploids or if it might apply more generally to outcrossed *S. pombe* strains. To address this, we compared fertility in the presence and absence of Rec12 in CBS5680/*Sp*, JB844/*Sp*, CBS5680/*Sk,* and JB844/*Sk* heterozygotes. We observed that Rec12 did not significantly promote fertility in heterozygotes as it did in homozygotes (*Figure 5C*). These results demonstrate that the meiosis fitness optimum in inbred strains differs from what is optimal when strains outcross.

We reasoned that competing *wtf* meiotic drivers were contributing to the dispensability of *rec12* in the outcrossed diploids. To test that idea, we assayed the fitness costs of deleting *rec12* in strains with heterozygous *wtf* drivers at one or two loci. We found that in a diploid with one set of heterozygous drivers (*Sk wtf4/Sk wtf28* at *ade6*), the cost of deleting *rec12* (*rec12Δ/rec12Δ*) was similar to that observed in the wild-type background (3-fold decrease in fertility) (*Figure 6—figure supplement 1*). However, we found that deleting *rec12* in a genetic background with *wtf* drivers competing at both *ade6* and *ura4* had no cost (*Figure 2C*, compare diploid 15 to diploid 13). These results support our model that the costs of disrupting chromosome segregation can be offset by the fitness benefits of disomic spores in the presence of *wtf* driver competition.

We next tested the fitness costs of deleting other genes that promote accurate meiotic chromosome segregation (*rec10*, *sgo1*, *moa1,* and *rec8*) in the presence and absence of competing *wtf* drivers (at *ade6*). Rec10 is a component of the meiotic chromosome axis (linear elements) that is required for the formation of most meiotic DSBs (*Lorenz et al., 2004*; *Prieler et al., 2005*). The Moa1 protein promotes monopolar attachment of sister kinetochores to ensure disjunction of homologs, rather than sister chromatids, in the first meiotic division (*Yokobayashi and Watanabe, 2005*). Sgo1 protects centromeric cohesion from cleavage during meiosis I and is required for proper segregation of sister chromatids during the second meiotic division (*Kitajima et al., 2004*). Finally, Rec8 is the meiotic kleisin that plays key roles in recombination and ensuring proper chromosome segregation in both meiotic divisions (*Krawchuk et al., 1999*; *Watanabe and Nurse, 1999*; *Yoon et al., 2016*).

Deleting *sgo1* and *rec10* had a lower fitness cost in the background with heterozygous *wtf* drivers than in the background without *wtf* competition (~3 fold decrease compared to a 6–7-fold decrease) (*Figure 6—figure supplement 1* and *Figure 6—figure supplement 2*). Remarkably, deleting *moa1*

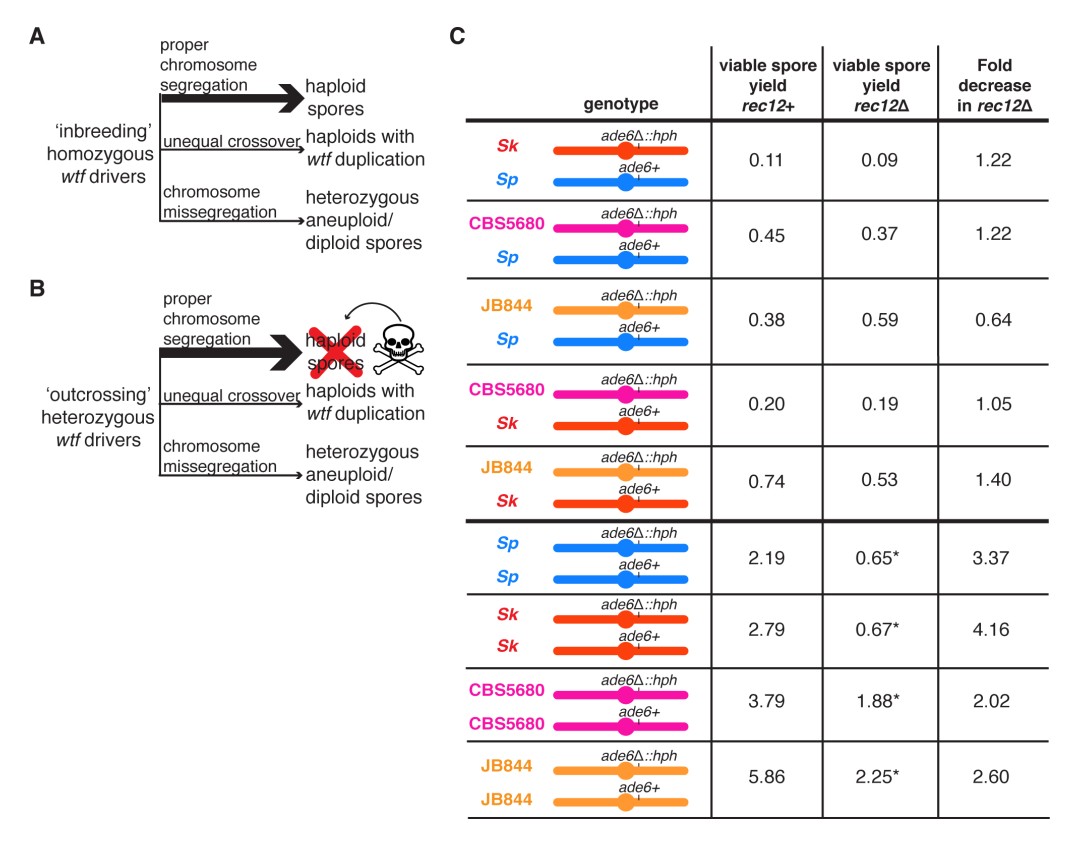

**Figure 5.** The Rec12 protein does not promote fertility in heterozygous *S. pombe* diploids. (**A**) Schematic of viable spores resulting from an 'inbreeding' scenario. The three indicated types of spores are viable when *wtf* drivers are homozygous. (**B**) Schematic of viable spores resulting from an 'outcrossing' scenario where one set of *wtf* drivers is heterozygous. Haploid spores that only inherit one *wtf* allele will be killed by the poison of the *wtf* they did not inherit. Spores that inherit both *wtf* drivers due to a *wtf* duplication or disomy (aneuploidy or diploidy) will survive. Other outcomes of meiosis are not represented in this figure. (**C**) Viable spore yield values of heterozygous and homozygous *S. pombe* diploids in *rec12+* and *rec12Δ* backgrounds. * indicates p-value<0.05 (Wilcoxon test) when comparing the *rec12+* to *rec12Δ* fertility values. We compared the viable spore yield of each diploid in *rec12+* and *rec12Δ* backgrounds. At least three, but usually more independent diploids were used to calculate viable spore yield. The data for *rec12+* diploids is repeated from *Figure 1*. The data for *Sk/Sk rec12Δ* diploid (diploid 19) is repeated from *Figure 3*. The raw data are reported in *Figure 5—source data 1*.

The online version of this article includes the following source data for figure 5:

**Source data 1.** Raw data of the viable spore yield reported in *Figure 5*.

or *rec8* had no effect on fitness in diploids with heterozygous *wtf* drivers, despite the fact that these mutations decrease fertility by 4- and 6-fold, respectively, in the absence of *wtf* driver competition (*Figure 6—figure supplement 1* and *Figure 6—figure supplement 2*).

We reasoned that heterozygous *moa1* (*moa1Δ/moa1+*) or *rec8* (*rec8Δ/rec8+*) mutants might slightly increase meiotic chromosome missegregation and thus provide a selective advantage in the presence of *wtf* competition (*Sk wtf4/Sk wtf28* at *ade6*) by producing disomic spores. Deleting one copy of *moa1* did not significantly alter the frequency of disomic spores in the absence of *wtf* driver competition (*Figure 6B*, diploid 28). In addition, heterozygosity for *moa1* did not suppress the fitness costs of *wtf* competition (*Figure 6B*, diploid 30). However, deleting one copy of *rec8* significantly increased the production of disomic spores in a background without heterozygous *wtf* drivers (*Figure 6B*, diploid 27). This suggests that *rec8* exhibits haploinsufficiency and reducing Rec8 protein levels may lead to chromosome segregation errors during meiosis. Consistent with this observation, mutations that reduce the levels of the meiotic cohesin can lead to meiotic defects in mice and flies (*Murdoch et al., 2013*; *Subramanian and Bickel, 2008*). As hypothesized, *rec8* heterozygosity also increased the fertility of diploids with competing drivers (*Figure 6B*, diploid 29). Overall, these

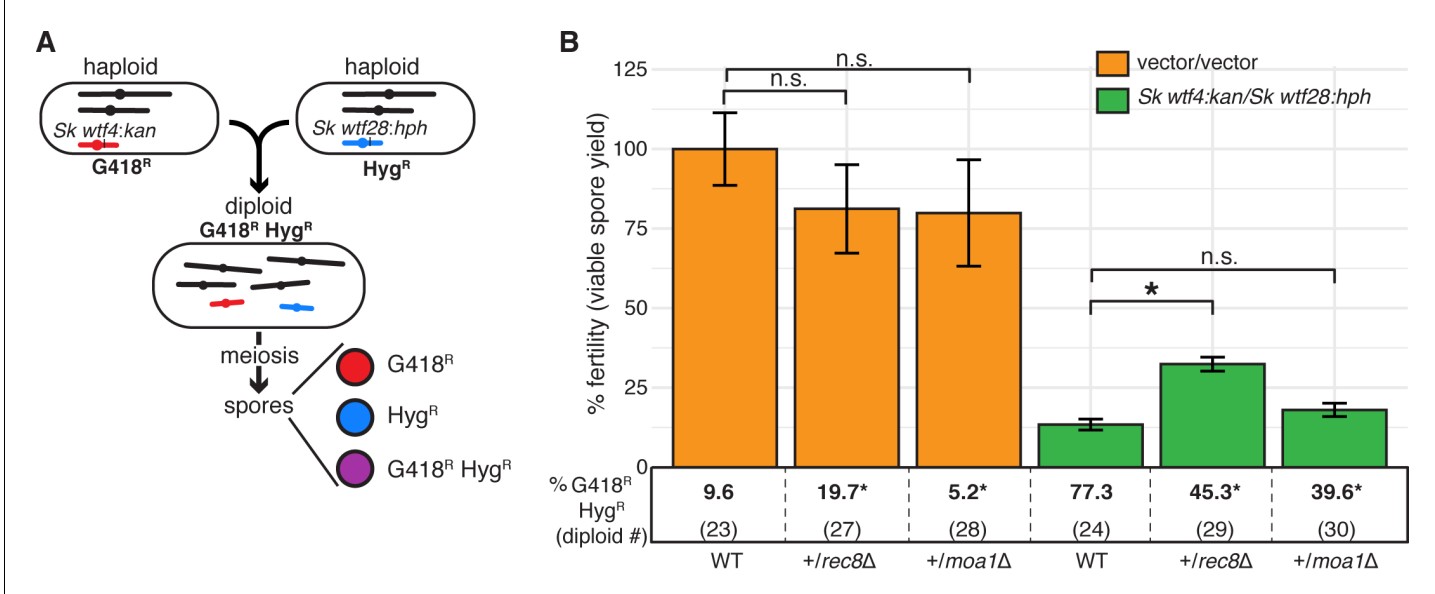

**Figure 6.** A heterozygous *rec8* mutation increases fitness when *wtf* meiotic drivers are in competition. (**A**) Schematic of diploid heterozygous for the Hyg[R] and G418[R] markers at the *ade6* locus. (**B**) Fertility was measured using the viable spore yield assay in diploids with markers linked to competing *wtf* drivers (*Sk wtf4/Sk wtf28*) or empty vectors (vector/vector). Error bars represent the standard error of the mean. Underneath each bar graph is the % of G418[R] Hyg[R] (aneuploid, diploid, or haploid with duplication event) spores for each diploid. * indicates p-value<0.05 (G-test [G418[R] Hyg[R]] and Wilcoxon test [fertility]). For statistical analyses, diploids 27 and 28 were compared to diploid 23, and diploids 29 and 30 were compared to diploid 24. Data for diploids 23 and 24 are repeated from *Figure 4*. Raw data are found in *Figure 6—source data 1* and *Figure 6—source data 2*.

The online version of this article includes the following source data and figure supplement(s) for figure 6:

**Source data 1.** Raw data of allele transmission values reported in *Figure 6* and *Figure 6—figure supplement 1*.
**Source data 2.** Raw data of the viable spore yield reported in *Figure 6* and *Figure 6—figure supplement 1*.
**Figure supplement 1.** Fitness costs of some meiotic mutants are reduced in diploids with competing *wtf* meiotic drivers relative to a background without *wtf* competition.
**Figure supplement 2.** Pictorial description of mutant effects on fertility and the frequency of G418[R] Hyg[R] spores.

results suggest that the costs of disrupting chromosome segregation can be partially or totally alleviated by the increased protection against *wtf* drivers gained by generating more heterozygous disomic spores.

## Driver competition can facilitate the maintenance or spread of alleles that disrupt meiotic chromosome segregation fidelity in a population

Our experiments demonstrate that the effects of meiotic mutants can be quite different in heterozygous *S. pombe* wherein *wtf* drivers are competing. To explore this idea further, we turned to population genetic modeling to analyze how drivers affect the evolution of variants that decrease the fidelity of meiotic chromosome segregation.

Our model analyzes the evolutionary fate of a hypothetical mutation that disrupts the segregation of chromosome 3, which houses the majority of *wtf* drivers. For the sake of simplicity, our model assumes that chromosome 3 exhibits whole-chromosome drive. The model also considers six parameters (*Figure 7A*). The first two parameters relate to the *wtf* drivers. We varied the number of driving alleles in the population ($n$) and the strength of their drive ($t$). Each driving allele was assumed to be at an equal frequency in the population and have the same strength of drive. The next parameters relate to the meiotic mutation. We varied the level of chromosome missegregation caused by the mutation ($f$) from 0 (no mutant phenotype) to 1 (50% of the resultant spores are heterozygous disomes and the remaining 50% of the spores lack chromosome 3 and are thus inviable). We considered the dominance of the mutation ($h$) and any additional fitness costs ($s_m$) the mutation may incur, such as potential costs relating to the missegregation of other chromosomes. Finally, we considered

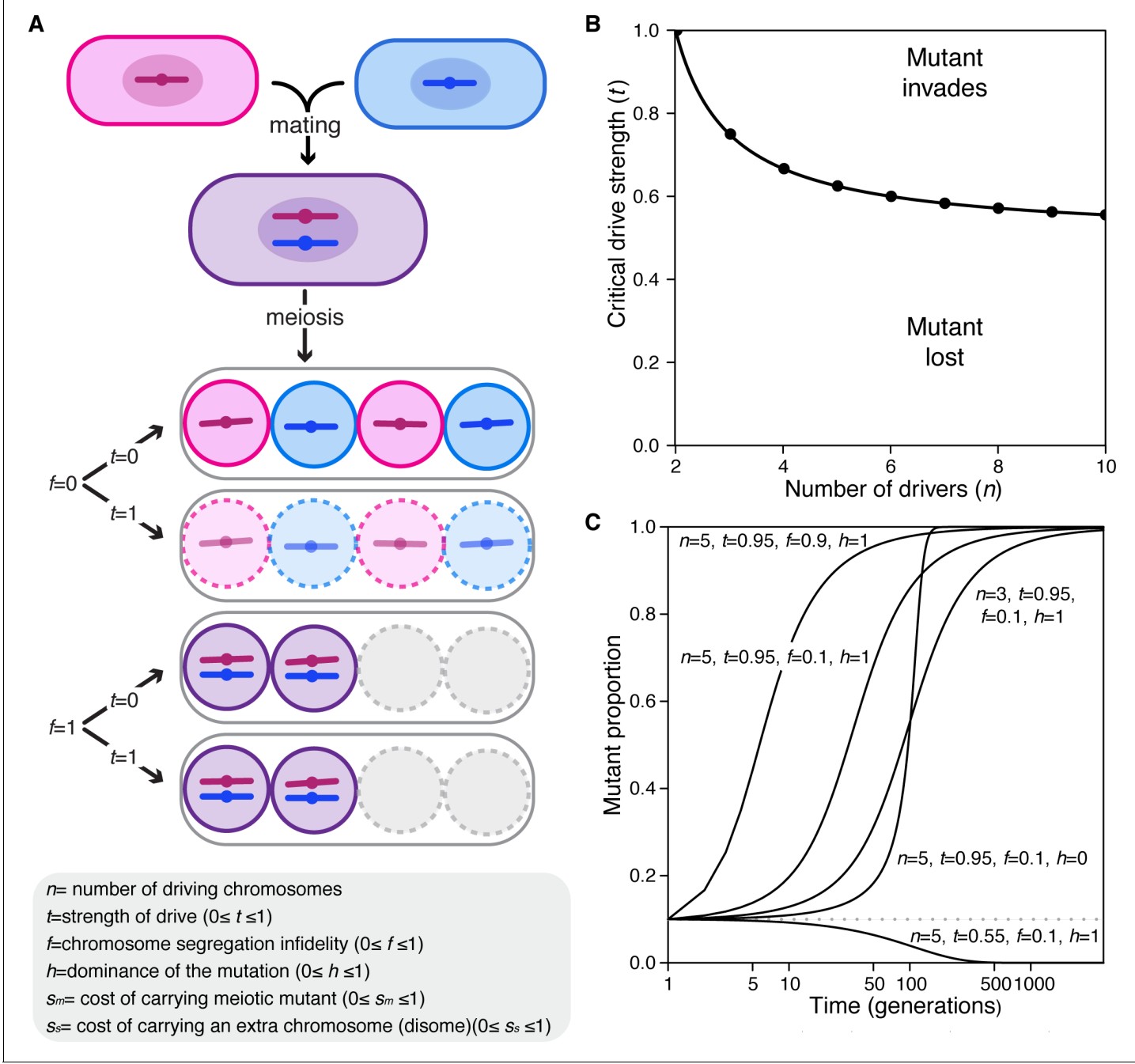

**Figure 7.** Population genetics of a meiotic mutant in response to meiotic drive. (**A**) Schematic of the spore progeny generated by *S. pombe* diploids when the infidelity of chromosome segregation (f) is 0 or 1 and the strength of drive (t) is 0 or 1. (**B**) Critical drive strength (t$_{critical}$) for invasion of the segregation infidelity mutant given *n* drivers. This assumes that there is no cost of the meiotic mutant. (**C**) Trajectories of meiotic mutants starting at a frequency of 0.1. Note that the X-axis is a log scale.

additional fitness costs (s$_s$) disomic spores might bear. The full description of the model and additional analyses are presented in Appendix 1.

We found that a mutation with no additional fitness costs (s$_m$ and s$_s$ = 0) that disrupts meiotic segregation could invade a population when:

$$t_{critical} > \frac{n}{2(n-1)},$$

where $t_{critical}$ is the value of drive strength necessary for such invasion in a population of $n$ drivers. Interestingly, the higher the strength of drive ($t$), the lower the number of drivers required for mutant invasion (*Figure 7B*). Importantly, our empirical work demonstrates that drive strength is generally high ($t > 0.9$) and that there are ample distinct *wtf* driver haplotypes (*Eickbush et al., 2019*) leaving parameter space for mutants to invade even if they incur considerable costs ($s_m$ and $s_s$) (*Appendix 1—figure 1*; *Bravo Núñez et al., 2018*; *Bravo Núñez et al., 2020*; *Eickbush et al., 2019*; *Hu et al., 2017*; *Nuckolls et al., 2017*). With fitness costs applied to the equation above, increasing the number of drivers in the population and decreasing the value of these associated costs both increase the likelihood that the mutation can invade the population (Appendix 1, *Appendix 1—figure 1*).

To get a broader perspective on the potential evolutionary trajectories of the segregation fidelity mutant, we varied the described parameters and plotted the results. These analyses all started with the mutant at a frequency of 0.1 in the population. We found that changes in some parameters dramatically influenced the trajectories. For example, with all other parameters fixed, at $t = 0.55$ the mutant is lost, while at $t = 0.95$, the mutant is fixed in the population (*Figure 7C*). With the parameters plotted, the dominance of the mutation, $h$, has little influence on the fate of the mutation but does influence the rate at which that fate is reached. Finally, in our model where the costs are applied, the cost of the mutation, the cost bore by disomic spores, and the degree of segregation infidelity also influence whether or not invasion occurs (*Appendix 1—figure 1*).

Overall, the results of the model are consistent with our experimental results. Both types of analyses support the idea that meiotic drivers can change the selective landscape of meiosis in outcrossed *S. pombe*. Instead of meiotic mutants being removed by negative selection due to fitness costs, variants that decrease the fidelity of chromosome segregation can be advantageous due to pervasive meiotic drive.

## Discussion

### Meiotic drivers shape the evolution of *S. pombe* meiosis

Most genetic analyses of meiosis are performed in inbred (homozygous) organisms. This approach has been incredibly powerful. It enabled Mendel to establish the founding principles of genetics with his true-breeding peas and facilitated countless discoveries over the last 150 years (*Abbott and Fairbanks, 2016*). However, studying inbred model systems with very low genetic variability has limitations in that phenotypes that can only be observed in heterozygotes remain hidden. These phenotypes can include those caused by selfish genetic elements, like meiotic drivers, that act primarily in heterozygous or outcrossed organisms. This work highlights the stark differences in fitness optima that can exist between gametogenesis in inbred homozygotes and in heterozygotes generated by outcrossing.

When *S. pombe* inbreeds to generate homozygotes, the effects of *wtf* meiotic drivers are largely invisible as every spore inherits the necessary Wtf antidotes required to neutralize the Wtf poisons. Granted these homozygotes faithfully segregate their chromosomes during meiosis and generate four haploid spores, the fitness of these diploids will be relatively high. However, mutations such as *rec12Δ*, *rec8Δ*, or *moa1Δ* will decrease the relative fitness of such diploids, as these mutations decrease the fidelity of chromosome segregation and spore viability (*Davis and Smith, 2003*; *Martín-Castellanos et al., 2005*; *Yokobayashi and Watanabe, 2005*).

When *S. pombe* outcrosses, the observed outcomes of meiosis are different from those observed under inbreeding, at least in part due to *wtf* meiotic drivers. These differences include changes in fertility, spore ploidy, recombination frequencies, and the fitness costs (or benefits) of inaccurate meiotic chromosome segregation. It is important to stress that the known effects *wtf* drivers exert on meiosis appear to be entirely indirect. There is no evidence that Wtf proteins directly participate in or affect the molecular mechanisms of recombination or chromosome segregation. Rather, the effects of the Wtf proteins (i.e. spore killing) are observed after the completion of meiosis. Despite this, the Wtf proteins have the power to indirectly affect the molecular steps of meiosis as changes in the fidelity of chromosome segregation or other aspects of gametogenesis can alter the number of spores destroyed by drive.

The *S. pombe* isolates in which *wtf* genes have been assembled carry between 4–14 intact *wtf* drive genes (*Eickbush et al., 2019*; *Hu et al., 2017*). The *wtf* genes are amongst the most rapidly evolving genes in *S. pombe*, which means outcrossing often generates extensive *wtf* driver heterozygosity. This leads to extensive death of haploid spores in outcrossed strains because 1) Wtf$^{antidote}$ proteins appear to neutralize only the Wtf$^{poison}$ proteins with highly similar or identical C-termini and 2) it is unlikely that haploid spores will inherit all drivers and thus encode all Wtf$^{antidote}$ proteins (*Bravo Núñez et al., 2018*; *Bravo Núñez et al., 2020*). Hence, generating the maximal number of haploid spores does not maximize the fitness of outcrossed *S. pombe*.

Instead, fitness is maximized in outcrossed *S. pombe* when the spores inherit as many drivers as possible. If there is just one locus with heterozygous drivers, perhaps due to mating between two closely-related isolates, an unequal crossover event can place both drivers on the same haplotype and allow a haploid spore to survive. It is possible this type of selection may be occurring in nature, as all assayed strains contain multiple loci with 2–3 *wtf* genes in tandem that could have been produced by unequal crossover events (*Eickbush et al., 2019*).

When two diverse isolates mate to form heterozygotes, the high number of *wtf* drivers present in each respective strain will make it unlikely for a haploid spore to inherit every driver. In this scenario, disomic spores that inherit two different copies of chromosome 3, which carries nearly every *wtf* gene, are most likely to inherit every driver and survive. Importantly, an extra copy of chromosome 3 is the only aneuploidy tolerated in *S. pombe* (*Niwa et al., 2006*).

We have previously speculated that the *wtf* gene family specifically expanded on chromosome 3 and not on chromosomes 1 or 2 as aneuploid spores provide an avenue to mitigate the fitness costs of multiple drivers (*López Hernández and Zanders, 2018*). The results of this study support and expand on that model. Specifically, we now show that drivers can create a selective landscape wherein variants that decrease the fidelity of chromosome segregation and thus generate more disomic spores can be favorable. In these cases, the fitness costs of mutating genes like *rec12*, *moa1,* or *rec8* can be offset by the fitness benefits of increased disomy. Our work also adds to previous work demonstrating meiotic drive-independent adaptive potential of aneuploid spores in other fungi (*Chuang et al., 2015*; *Ni et al., 2013*).

It is not clear how often *S. pombe* outcrosses in the wild, and population genetics estimates are confounded by drive and repressed recombination in hybrids (*Farlow et al., 2015*; *Fawcett et al., 2014*; *Jeffares et al., 2015*; *Tusso et al., 2019*; *Zanders et al., 2014*). Many *S. pombe* isolates can switch mating type during clonal growth and thus mate with nearby clonal cells when starved for nutrients (*Egel, 1977*). This undoubtedly leads to frequent inbreeding in *S. pombe* and could thus promote selection against mutations that increase the frequency of disomic spores. However, when outcrossing occurs, mutations that increase disomy can have a selective advantage. A mix of inbreeding/outcrossing strategies could lead to the maintenance of variation in the frequency at which meiosis generates disomic spores. Consistent with this, we observed such variation amongst the natural isolates assayed in this study (*Figure 1C*, diploids 8–12 and *Figure 1—figure supplement 3*). Strikingly, the strains with the 'highest' meiotic fidelity still make ~5% disomic spores, suggesting that chromosome 3 missegregates during the first meiotic division in one out of ten meioses.

Given the results of this study, it is difficult to resist speculating about the potential evolutionary relationships between the frequent disomy of *S. pombe* spores and the *wtf* drivers. It is possible that frequent disomy of *S. pombe* spores was selected to mitigate the cost of *wtf* drivers. An alternative model is that frequent disomy perhaps predates and/or contributes to the success of the *wtf* drivers. In this model, the disomy could be beneficial to the *wtf* genes by preventing them from being too costly to their host and thereby destroying themselves by destroying the host. Unfortunately, at this time we know far too little about fission yeast ecology and evolution to rigorously explore these possibilities (*Jeffares, 2018*). More knowledge about population structures, outcrossing frequencies, meiotic driver prevalence, and disomy frequencies in other fission yeast species is required to distinguish between the proposed models.

## The effects of drive on the evolution of gametogenesis outside of *S. pombe*

A growing body of evidence indicates that meiotic drive is pervasive in eukaryotes, and more drivers are identified each year. This includes the gamete-killing type of meiotic drivers described in this

work (*Bauer et al., 2012*; *Bravo Núñez et al., 2018*; *Bravo Núñez et al., 2020*; *Burt and Trivers, 2006*; *Didion et al., 2015*; *Grognet et al., 2014*; *Hammond et al., 2012*; *Hu et al., 2017*; *Larracuente and Presgraves, 2012*; *Long et al., 2008*; *Nuckolls et al., 2017*; *Pieper et al., 2018*; *Rhoades et al., 2019*; *Vogan et al., 2019*; *Xie et al., 2019*; *Yang et al., 2012*; *Yu et al., 2018*), but also extends to other drivers that use completely different methods to gain a transmission advantage. For example, biased gene conversion favoring unbroken DNA during meiotic recombination is a form of meiotic drive tied to the mechanisms of DSB repair (*Marais, 2003*). This type of drive shapes recombination landscapes and likely promotes the rapid evolution of at least one key recombination protein found in many mammals, including humans (*Grey et al., 2018*; *Úbeda et al., 2019*). Other meiotic drivers exploit the asymmetry of female meiosis to promote their transmission into the one viable meiotic product (i.e. the oocyte) (*Akera et al., 2017*; *Akera et al., 2019*; *Dawe et al., 2018*; *Kato Yamakake, 1976*; *Rhoades, 1942*). This type of bias has been hypothesized to drive the widespread rapid evolution of karyotypes, centromere sequences, and centromeric proteins (*Pardo-Manuel de Villena and Sapienza, 2001*; *Henikoff et al., 2001*; *Rosin and Mellone, 2017*). In addition, drive during female meiosis in mice can even generate selective pressure to alter the timing of the first meiotic division (*Akera et al., 2017*; *Akera et al., 2019*).

It may be tempting to disregard the *wtf* genes within *S. pombe* as an anomaly. However, meiotic drivers are ubiquitous, and drive represents an incredibly powerful evolutionary force (*Burt and Trivers, 2006*; *Sandler and Novitski, 1957*). Appreciating how *wtf* genes affect *S. pombe* will likely provide important insights into how genetic parasites can shape the evolution of meiosis in other eukaryotes.

## Materials and methods

### Strain construction: *S. pombe* natural isolates

All yeast strain names and genotypes are described in *Supplementary file 1*. We made the *lys1Δ::kanMX4* and *ade6Δ::hphMX6* alleles used in *Figure 1* as described in *Zanders et al., 2014*. Using the standard lithium acetate protocol, we independently transformed the cassettes into seven different *S. pombe* natural isolates (JB844, JB1172, CBS5680, JB873, JB939, JB929, and NBRC0365). However, we were only successful at transforming both markers into JB844, JB1172, CBS5680, and NBRC0365. We could not find conditions in which to mate and sporulate NBRC0365.

To generate a *rec12Δ::ura4+* deletion in the CBS5680 strain background, we first made a *ura4-D18* mutation in SZY2111 (*ade6Δ::hphMX6* in CBS5680). We amplified the *ura4-D18* allele from SZY925 using oligos 35 and 38 and transformed it into SZY2111 to generate SZY3949. We then amplified the *rec12Δ::ura4+* cassette from SZY580 using oligos 1194 and 1077 and transformed the cassette into SZY3949 to generate SZY3995. We confirmed the *rec12* deletion via PCR using oligos (1120+1108) that bind 730 bases upstream and 224 bases downstream of the deletion cassette. We generated the *rec12Δ::ura4+* deletion in the *lys1Δ::kanMX4* background of the CBS5680 isolate via crosses. We generated the *rec12* deletion in JB844, similarly to how we generated it in the CBS5680 strain.

We found it difficult to make gene deletions in many of the natural isolates used in this study. We had more success, however, making mutations using integrating vectors. Because of this, we used integrating vectors to generate the genetic markers used in *Figure 1—figure supplement 3*. We used pSZB386 to generate haploid strains with a *hphMX6* marker at *ade6*, without deleting the *ade6* gene. We digested this plasmid with KpnI and transformed it into different natural isolates, selecting for transformants that were resistant to Hygromycin B and appeared red on media with low adenine (*Bravo Núñez et al., 2018*). To generate strains with a *kanMX4* marker at *lys1*, we first ordered a gBlock from IDT (Coralville, IA). This gBlock contained ~1000 bp from the middle of the *lys1* gene in which we replaced 50 bp from the center with a KpnI site. We then cloned the gBlock into the BamHI and SalI sites of pFA6 to generate pSZB816. We then digested pSZB816 with KpnI and transformed it into different *S. pombe* isolates. Finally, we screened for transformants that grew on plates containing G418 and were not able to grow on media lacking lysine.

## Plasmid construction: integrating vectors with *wtf* alleles

Most of the integrating vectors containing *wtf* alleles were previously described in *Bravo Núñez et al., 2018*; *Bravo Núñez et al., 2020*; and *Nuckolls et al., 2017*. To generate the additional *ade6-* and *ura4*-integrating vectors unique to this work, we cloned the *wtf* genes of interest into the integrating plasmid backbones and confirmed them via sequencing. The DNA templates, oligos, and restriction enzymes used are described in *Supplementary file 3*.

To generate pSZB923 (*Sk wtf4:natMX4*), we first digested pSZB189 (which contains *Sk wtf4*) with SacI to release the *Sk wtf4* cassette. We then cloned the cassette into the SacI site of pSZB849.

To make the *Sk wtf28* poison-only allele, we mutated the two start sites found in exon one to stop codons (ATG→TAC). Using pSZB254 as a template, we amplified the 5' fragment (with oligos 651+954) and the 3' fragment (with oligos 953+733) of *Sk wtf28*. We used overlap PCR to stitch the fragments together and digested them with SacI. We then cloned this fragment into the SacI site of pSZB386 to generate pSZB414.

## Deletions of the *moa1, rec10,* and *sgo1* genes in *Sp*

We made *moa1*, *rec10*, and *sgo1* gene deletions using standard deletion cassettes and transformation. To make the *moa1Δ::natMX4* cassette, we amplified the upstream region of *moa1* with oligos 1673+1187 and the downstream region with oligos 1190+1191 (or 1190+1674) using SZY643 as a template. We also amplified the *natMX4* gene (with oligos 1675+1189) using pAG25 as a template (*Goldstein and McCusker, 1999*). Next, we stitched all the PCR fragments together using overlap PCR and transformed this fragment into SZY44 and SZY643 to make strains SZY2479 and SZY2481, respectively. We confirmed the integration of the deletion cassette at the *moa1* locus using oligos AO638+1192, AO1112+1191, and 1701+1702. We also checked that the *moa1* gene was not present somewhere else in the genome by using two oligos (1703+1704) within *moa1*.

To generate a *rec10Δ::natMX4* strain, we first amplified the upstream region and the downstream region of *rec10* from SZY643 using oligos 1723+1724 and oligos 1727+1728, respectively. We also amplified the *natMX4* cassette from pAG25 using oligos 1725+1726 (*Goldstein and McCusker, 1999*). Using overlap PCR, we stitched the three PCR fragments together and then transformed the final deletion cassette into SZY643 and SZY44 to make strains SZY2517 and SZY2519, respectively. To confirm the integration of the cassette at the correct locus, we used oligo pairs 1731+AO638 and 1732+AO1112. We also confirmed the absence of the wild-type *rec10* gene by using internal oligos (1729+1730).

To make the *sgo1Δ::hphMX6* allele, we amplified the sequences upstream and downstream of *sgo1* from SZY643 using oligos 1224+1225 and 1228+1229. We also amplified the *hphMX6* cassette from pAG32 using oligos 1226+1227 (*Goldstein and McCusker, 1999*). We then stitched all the PCR fragments together using overlap PCR and transformed the cassette into yeast to generate strains SZY1735 and SZY1736. We confirmed the *sgo1* deletion using oligos AO638+1230, AO1112 +1231, and 1230+1231. We then confirmed the absence of the wild-type gene using an internal oligo pair (2088+2089).

### *Sp wtf4* deletion

To delete *Sp wtf4,* we utilized the CRISPR/Cas9-based method described in *Rodriguez-Lopez et al., 2016*. We first cloned a plasmid (pSZB570) encoding Cas9 and a guide RNA targeting *Sp wtf4*. To do that we first amplified pMZ379 (plasmid containing Cas9) using oligos 1206+1207. These oligos contained the single guide RNA (sgRNA) sequence that targets the *Sp wtf4* gene. We then ligated the ends together to generate pSZB570.

We also made a deletion cassette to knockout the *Sp wtf3* and *Sp wtf4* locus. We used oligos 574 +1138 and 1139+471 to amplify the upstream and downstream sequence of the locus using SZY580 as a template. We stitched these two fragments together using overlap PCR. Next, we transformed the deletion fragment and pSZB570 into SZY1595 to generate SZY1699. We used oligos 1069+543 to confirm the *Sp wtf4* deletion. We also Sanger sequenced the PCR fragment and found that we had only knocked out the *Sp wtf4* gene, not the entire *Sp wtf3* and *Sp wtf4* locus.

### *Sp wtf13* deletions

The *Sp wtf13* deletions were made similar to the ones described in *Bravo Núñez et al., 2018*. Using SZY580 as a template, we amplified the upstream (with oligos 1048+1049) and downstream (with oligos 1052+1053) sequence of *Sp wtf13*. Additionally, we amplified the *kanMX4* cassette using oligos 1050+1051, with pFA6 as a template (*Wach et al., 1994*). We then stitched the upstream region, *kanMX4*, and the downstream region together using overlap PCR to make an *Sp wtf13Δ::kanMX4* cassette. After, we transformed this fragment into SZY580 to generate SZY1391-SZY1394.

To generate the *Sp wtf13Δ::kanMX4 Sp wtf4Δ* strain, we first digested pAG25 with EcoRV and BamHI to release the *natMX4* cassette (*Goldstein and McCusker, 1999*). We then used this cassette to switch the *kanMX4* marker (at the *his5* gene) from the SZY1699 strain via transformation to generate SZY1981 and SZY1982. We transformed the *Sp wtf13Δ::kanMX4* deletion cassette into SZY1981 and SZY1982 to generate strains SZY2008 and SZY2010, respectively. We confirmed these deletions using a series of PCR reactions. We used two oligo pairs with one oligo outside of the deletion cassette and one oligo internal to the *Sp wtf13* gene (1058+1059 and 1060+1061) and two oligo pairs in which one oligo was external to and one oligo was within the deletion cassette (1058+AO638 and 1061+AO112).

### Fertility and allele transmission

We assayed fertility and allele transmission as described in *Bravo Núñez et al., 2020*.

Some of the spore colonies from *ade6Δ::hphMX6/ade6+; lys1+/lys1Δ::kanMX4* CBS5680 diploids (SZY2213/SZY2111) were small and red. When we determined their genotype, the colonies were adenine auxotrophs and took five days to grow when replicated to fresh media. We supplemented the plates with more adenine, but the colonies did not grow faster. This slow growth phenotype was curiously not observed in the Ade- parental haploid (SZY2111).

### Recombination frequency within the *ade6* and *ura4* interval

To determine the recombination frequency for diploids 20 and 21, we needed to distinguish the *ura4* allele (*ura4-294* or *ura4Δ::kanMX4*) via PCR. We amplified the *ura4* locus using two sets of oligos (34+37 and 34+AO638).

For *Figure 2—figure supplement 5* and *Figure 3—figure supplement 2*, we performed power analyses to estimate the total number of spores we needed to assay in order to determine an effect in recombination frequencies (alpha = 0.05 and power = 80%).

### Determining ploidy of spore colonies

In the various tests to assay the ploidy of the spore colonies for *Figure 4*, we compared the spore colony phenotypes to the following control strains: a homothallic haploid (SZY925), a heterothallic haploid (SZY1180), a diploid (SZY925/SZY1180), and aneuploid (irregular colonies generated by a cross between SZY1994 and SZY1770) controls. The ploidy of the strains was determined by how closely a test strain resembled one of the controls in the following tests:

#### Spore colony morphology

To determine the morphology of the spore colonies from different diploids, we diluted the spores to get isolated colonies on YEA+S (0.5% yeast extract, 3% dextrose, agar, and 250 mg/L adenine, histidine, uracil, leucine, and lysine). We then imaged the colonies using the Canon EOS Rebel T3i plate imager. We marked each cell in ImageJ and determined the genotype of each spore colony via replica plating. These images allowed us to correlate the morphology of each spore colony with its genotype. For the spores that had resistance to both G418 and Hygromycin B, we assessed if the colonies were either large, medium, or small, and if the morphology was either round or irregular. Round colonies are typically haploid or diploid. Small and irregular-shaped colonies are characteristic of 'sick' colonies or aneuploids (*Niwa et al., 2006*).

#### Chromosome loss assay

To determine the ploidy of the spore colonies, we determined the frequency at which each of the G418$^R$ Hyg$^R$ strains lost one of the drug markers during vegetative growth. Aneuploids frequently lose their extra chromosome during vegetative growth and it is random which chromosome is lost

(*Niwa et al., 2006*). Diploids and haploids are expected to be more stable. We began by culturing ~26 G418$^R$ Hyg$^R$ spore colonies produced by each experimental diploid (diploids 23–26), along with six haploid controls (deemed haploid due to the presence of only one drug marker) in 5 mL of YEL (0.5% yeast extract, 3% dextrose, and 250 mg/L adenine, histidine, uracil, leucine, and lysine) with shaking at 32°C for 24 hr. The next day (day 1), we diluted the cultures and plated the cells on YEA to assay the presence or absence of the drug resistance markers. We also made 1:10 dilutions into 650 µL of YEL in 96-well plates and grew them for 24 hr. We repeated this for five days. Aneuploids readily and randomly lost one marker around day 1 or 2, so that a high fraction (17–100%) of the colonies generated by plating the culture were no longer resistant to both drugs. Haploids and diploids, however, maintained both markers for all 5 days because almost all (~90%) of the colonies generated by plating the culture were still resistant to both drugs. The marker maintenance value was calculated by dividing the number of G418$^R$ Hyg$^R$ colonies by the total number of colonies that grew on YEA+S.

## Phloxin B staining

Phloxin B was used to differentiate between diploid strains and haploids. Phloxin B is a dye that enters cells with compromised membranes, staining them red (*Forsburg and Rhind, 2006*). Diploids have a higher concentration of dead cells within a colony than haploids due to their lower stability resulting in a red stain. Haploids are much more stable, which reduces the ability of phloxin B to enter cells, leading haploids to appear white. Haploid strains that are homothallic (h$^{90}$) look pink (*Figure 4—figure supplement 3*). We spotted 10 µL of the saturated culture of day 1 (described above) onto YEA+S plates containing 5 mg/L phloxin B and grew the cells at 32°C overnight. We then determined if the spots were red, pink, or white by comparing them to the controls.

## Microscopy and iodine staining from SPA plates

Using the cultures from day 1 (described above), we also spotted 10 µL of the saturated culture of each strain and controls onto two SPA plates. We then placed these plates at 25°C for 20 hr. From the first SPA plate, we imaged the cells on a Zeiss (Germany) Observer Z.1 widefield microscope with a 40X (1.2 NA) water-immersion objective and acquired the images using the µManager software. Twenty hours was enough time for diploids to sporulate but not enough for homothallic haploid strains to mate, form diploids, and sporulate, allowing for further distinction between these ploidies. Although this assay did not reliably allow us to distinguish aneuploid cells that could not sporulate (as they would resemble the heterothallic haploid control), we were able to score some homothallic 'aneuploid' strains due to the presence of asci with an abnormal shape or number of spores.

The second SPA plate was stained with iodine (*Forsburg and Rhind, 2006*) after 20 hr at 25°C. Iodine vapors stain the starch present in spore walls a dark brown color, while heterothallic haploid cells that cannot sporulate appear yellow. Diploids stain dark brown, while homothallic haploids that first needed to mate in order to sporulate stained light brown (*Figure 4—figure supplement 3*).

## PCR assay for duplications

Some of the G418$^R$ Hyg$^R$ spores we tested appeared to be haploids based on the assays described above. We reasoned they could be the result of an unequal crossover putting both marker genes onto one chromosome. To test this, we used two sets of oligo pairs (2415+2417 and 2416+2418). These PCR reactions only work if the drug cassettes are found in tandem. To confirm that the presence of only one cassette would not lead to band amplification, we used the haploid parental strains (SZY925, SZY1180, SZY887, and SZY1293) as negative controls.

## **PI staining and confocal microscopy**

To determine the number of spores in asci that lost membrane integrity, we stained the spore sacs using the propidium iodide (PI) dye (1 mg/mL) (*Nuckolls et al., 2017*). These spores were generated by SZY2625/SZY1180, SZY2628/SZY1180, SZY1293/SZY2625, and SZY2628/SZY1293 diploids. We scraped the cells off of the SPA plates and swirled them into 50 µL of ddH$_2$0 in an Eppendorf tube. We then incubated the sample at room temperature for 20 min. After the incubation, we mixed the cells with 50 µL lectin (1 mg/mL) and plated them onto a 35 mm glass culture dish (MatTek) pre-

coated with lectin to immobilize the cells. We imaged the samples on an LSM-780 (Zeiss) AxioOb-server microscope with a 40X C-Apochromat water-immersion objective (NA = 1.2) on photon-counting channel mode with 561 nm excitation. We acquired the PI fluorescence through a 562–642 nm filter. We assayed more than 50 asci for each genotype.

## Acknowledgements

We thank Adèle Marston, Thomas Price, and members of the Zanders lab for their helpful comments that improved the manuscript. We are grateful to Gerry Smith for sharing the *rec8* mutant strain, and to Jeff Lange and Valeria Eliosa for technical support. This work was performed to fulfill, in part, requirements for MABN's thesis research in the Graduate School of the Stowers Institute. Original data underlying this manuscript can be accessed from the Stowers Original Data Repository at http://www.stowers.org/research/publications/libpb-1514. This work was supported by the following awards to SEZ: The Stowers Institute for Medical Research (https://www.stowers.org), the March of Dimes Foundation Basil O'Connor Starter Scholar Research Award No. 5-FY18-58 (https://www.marchofdimes.org), the Searle Scholar Award, and the National Institutes of Health (NIH) under the award numbers R00GM114436 and DP2GM132936 (https://www.nih.gov). MABN was also supported by the National Cancer Institute of the NIH under award number F99CA234523. RLU was supported by funding from the University of Kansas. The funders had no role in study design, data collection and analysis, or manuscript preparation.

## Additional information

### Competing interests

María Angélica Bravo Núñez, Sarah E Zanders: Inventor on a patent application based on *wtf* killers (patent application serial 62/491,107). The other authors declare that no competing interests exist.

### Funding

| Funder | Grant reference number | Author |
| --- | --- | --- |
| Stowers Institute for Medical Research | | Sarah E Zanders |
| March of Dimes Foundation | Basil O'Connor Starter Scholar Research Award No. 5-FY18-58 | Sarah E Zanders |
| Kinship Foundation | Searle Scholars Award | Sarah E Zanders |
| National Institute of General Medical Sciences | R00GM114436 | Sarah E Zanders |
| National Institute of General Medical Sciences | DP2GM132936 | Sarah E Zanders |
| National Cancer Institute | F99CA234523 | María Angélica Bravo Núñez |
| University of Kansas | | Robert L Unckless |

The funders had no role in study design, data collection and interpretation, or the decision to submit the work for publication.

### Author contributions

María Angélica Bravo Núñez, Conceptualization, Data curation, Formal analysis, Funding acquisition, Validation, Investigation, Visualization, Methodology, Writing - original draft, Writing - review and editing; Ibrahim M Sabbarini, Data curation, Formal analysis, Validation, Investigation, Visualization, Methodology, Writing - original draft, Writing - review and editing; Lauren E Eide, Data curation, Formal analysis, Validation, Investigation; Robert L Unckless, Conceptualization, Data curation, Formal analysis, Validation, Investigation, Visualization, Methodology, Writing - original draft, Writing - review and editing; Sarah E Zanders, Conceptualization, Data curation, Formal analysis, Supervision,

Funding acquisition, Visualization, Methodology, Writing - original draft, Project administration, Writing - review and editing

**Author ORCIDs**
María Angélica Bravo Núñez 🔾 https://orcid.org/0000-0002-6554-8814
Ibrahim M Sabbarini 🔾 http://orcid.org/0000-0001-6490-7056
Robert L Unckless 🔾 http://orcid.org/0000-0001-8586-7137
Sarah E Zanders 🔾 https://orcid.org/0000-0003-1867-986X

**Decision letter and Author response**
Decision letter https://doi.org/10.7554/eLife.57936.sa1
Author response https://doi.org/10.7554/eLife.57936.sa2

## Additional files

### Supplementary files
• Supplementary file 1. Yeast strains used in this study. #Strains SZY1535 and SZY1537 are Ura+ because when the *Sk wtf4* allele was originally made (described in *Nuckolls et al., 2017*), the strains retained the *ura4+* cassette. The *ura4* cassette is at an unknown location but linked to the *wtf4* locus. *Strains SZY3910 and SZY3911 were generated via crossing. One of the parental strains had a wild-type *ura4* allele and was thus Ura+. The other parental strain contained the *ura4-D18,* but had a *rec12Δ::ura4+*, and thus was also Ura+. However, when we crossed the strains to generate SZY3910 and SZY3911, we confirmed the presence of the *rec12* allele using PCR, but we did not determine if they contained the *ura4-D18* or the *ura4+* allele.

• Supplementary file 2. Plasmids used in this study.

• Supplementary file 3. Table summary of plasmid construction. Column 1 lists the *wtf* gene cloned into each vector. Column 2 denotes the isolate from which each *wtf* was cloned. The DNA templates and oligos used in the PCR reactions to amplify the *wtf* alleles are shown in columns 3 and 4, respectively. We digested each of the amplified fragments with the enzymes reported in column 5 and then integrated into the target site listed in column 6. The strain number of each of the plasmids that we generated is reported in column 7. The description of each plasmid can be found in *Supplementary file 2*.

• Supplementary file 4. Oligo table.

• Transparent reporting form

### Data availability
All data generated or analysed during this study are included in the manuscript and supporting files.

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

## Appendix 1

### Population genetics model

It is counterintuitive that a mutation causing reduced fidelity of chromosome segregation during meiosis could ever be beneficial and spread through a population. To determine whether such a segregation infidelity mutation could invade a population with *wtf*-like meiotic drivers, we created a simple mathematical model of the scenario.

### The model

As described in the main text, we start with a random mating population of infinite size with *n* distinct drivers – all at the same locus on the same homologous chromosome. Each driving chromosome has the same strength of drive (*t*) and is completely susceptible to all other driving chromosomes (e.g. the antidote of the 3$^{rd}$ driver does not protect against the poison of the 2$^{nd}$ driver). We assume that each drive chromosome is at equal frequency so the frequency of the *i*$^{th}$ driver is $P_i = 1/n$. This means that there are no chromosomes without a driver in the population – a reasonable assumption given observations in isolates of *S. pombe* (*Bravo Núñez et al., 2018*; *Bravo Núñez et al., 2020*; *Eickbush et al., 2019*; *Hu et al., 2017*; *Nuckolls et al., 2017*). With random mating, the proportion of the population that are homozygotes when it comes to the drive chromosome is $P_{homozygote} = n\left(\frac{1}{n}\right)^2 = \frac{1}{n}$ and the proportion that are heterozygotes is $P_{heterozygote} = 1 - n\left(\frac{1}{n}\right)^2 = 1 - \frac{1}{n}$. Since all chromosomes contain distinct *wtf* drive loci, all heterozygotes are likely to produce spores that will be killed by the Wtf$^{poison}$.

The strength of drive (*t*) ranges from zero (Wtf$^{poison}$ does not kill any spores) to one (Wtf$^{poison}$ kills all spores that do not produce the specific Wtf$^{antidote}$). Therefore, the relative number of spores produced by a heterozygous diploid is 1–*t* and the average relative fitness of a population with *n* drivers all at equal frequency ($P_i$) and strength of drive (*t*) is:

$$w_{drive\ population} = \frac{1}{n} + \left(1 - \frac{1}{n}\right)(1-t) = 1 - t + \frac{t}{n} \tag{S.1}$$

This means that a population with *n* drivers is, on average, (*t-t/n*) less fit than a population without drivers (*Appendix 1—figure 1A*). This 'drive load' is quite severe with large *t* and large *n*. Based on our findings, various isolates of *S. pombe* contain *wtf* drivers that exhibit a drive strength around *t* = 0.9 and different isolates contain multiple, diverse *wtf* drivers (this number varies considerably) (*Bravo Núñez et al., 2020*; *Eickbush et al., 2019*; *Hu et al., 2017*). For example, if a given population contained five segregating drivers, the fitness of this population would be 1–0.9 + (0.9/5)=0.28 (or 72% less fit) relative to a population without drive. Such a burden would select for resistance mechanisms that might include positive assortative mating or inbreeding, resistance alleles such as *wtf18-2* or aneuploidy (*Bravo Núñez et al., 2018*). With aneuploidy, some fraction of the spores (*f/2*, see below) would inherit both drive chromosomes, allowing the spores to produce both antidotes and be protected from the Wtf poisons. However, an equal fraction (*f/2*) would receive zero copies of chromosome 3 and would therefore be inviable (*Niwa et al., 2006*).

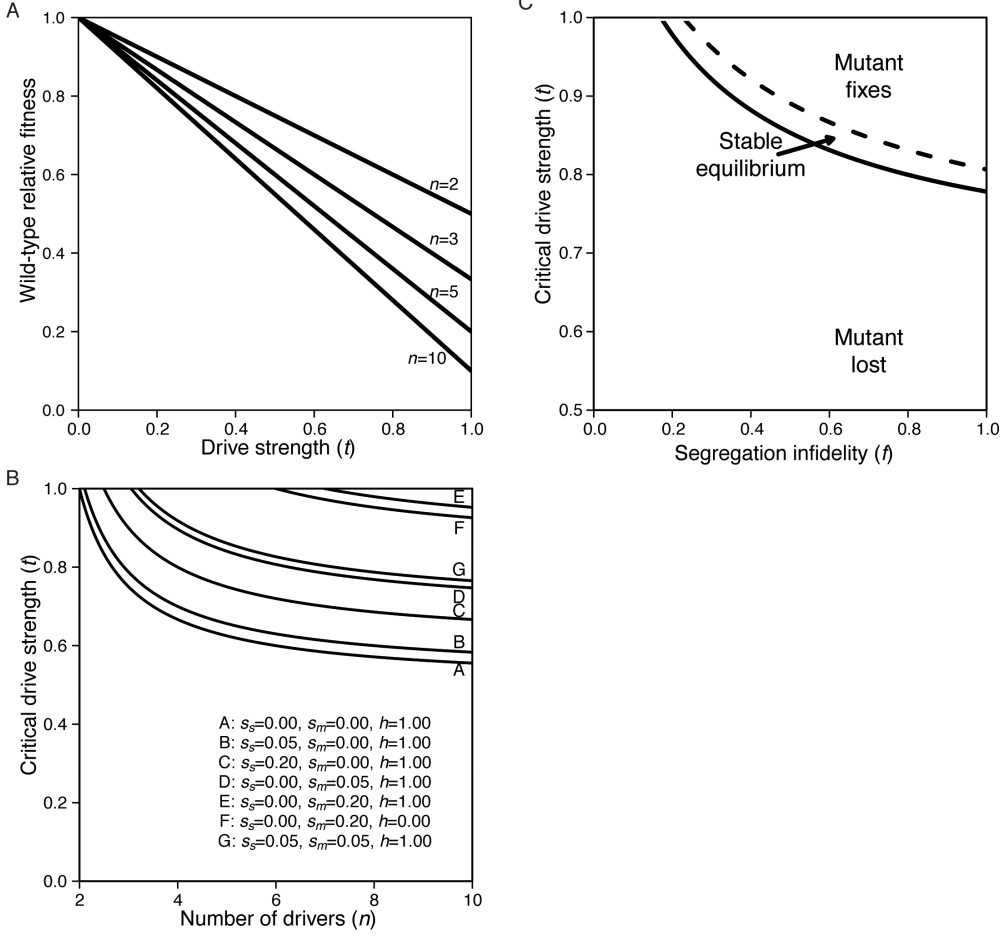

**Appendix 1—figure 1.** The drive load due to *wtf* segregating within populations. (**A**) From *Equation S.1*, $n$ is the number of drivers in the population and $t$ is the strength of drive. Wild-type fitness is relative to a population without any drivers. (**B**) Meiotic mutant invasion criteria with extrinsic costs. All curves assume $f = 0.1$. Other parameters are listed in the plot. A mutant is expected to invade when the parameters for $t$ and $n$ fall above the line of interest. (**C**) Narrow parameter space allows for stable equilibrium. Solid line represents $t_{critical}$, below which the segregation mutant cannot invade. Dashed line represents $t_{fixation}$, above which the mutant fixes. In the space between the lines, the equilibrium ranges from zero (solid line) to one (dashed line). $n = 5$, $s_m = 0.2$, $s_s = 0.1$, $h = 1$.

We assume that a single locus controls segregation fidelity and that a mutation at that locus (termed a segregation infidelity mutant) results in an increase in the number of aneuploid spores produced due to a defect in meiosis I. We also assume that this mutation only influences segregation of chromosome 3 (where most *wtf* drivers reside) but is not on the third chromosome itself (*Bowen et al., 2003*; *Eickbush et al., 2019*; *Hu et al., 2017*). However, reduced segregation fidelity of other chromosomes can be included in the costs discussed below. The parameter $f$ denotes the segregation infidelity and can range from 0 (all spores are haploid) to 1 (all meioses result in 50% of spores being disomic and 50% lacking a third chromosome). Therefore, this segregation infidelity mutant has an intrinsic fitness cost since 50% of the generated spores lack a third chromosome and are therefore inviable. The recursion for the frequency of the infidelity mutant is:

$$q' = \frac{1}{w}\Big(2q(1-q)\Big(\frac{1}{n}\Big(\frac{1-hf}{2}+\frac{hf}{4}\Big)+\Big(1-\frac{1}{n}\Big)\Big((1-t)\Big(\frac{1-hf}{2}+\frac{hf}{4}\Big)+t\Big(\frac{hf}{4}\Big)\Big)\Big)$$
$$+q^2\Big(\frac{1}{n}\Big(1-f+\frac{f}{2}\Big)+\Big(1-\frac{1}{n}\Big)\Big((1-t)\Big(1-f+\frac{f}{2}\Big)+t\frac{f}{2}\Big)\Big)\Big) \tag{S.2A}$$

$$\bar{w} = (1-q)^2\left(\tfrac{1}{n} + \left(1-\tfrac{1}{n}\right)(1-t)\right) + 2q(1-q)\left(\tfrac{1}{n}\left(1-hf+\tfrac{hf}{2}\right) + \left(1-\tfrac{1}{n}\right)\right.$$
$$\left(\left((1-t)\left(1-hf+\tfrac{hf}{2}\right)+t\left(\tfrac{hf}{2}\right)\right)\right)\right) + q^2\left(\tfrac{1}{n}\left(1-f+\tfrac{f}{2}\right)+\right. \quad \text{(S.2B)}$$
$$\left(1-\tfrac{1}{n}\right)\left((1-t)\left(1-f+\tfrac{f}{2}\right)+t\tfrac{f}{2}\right)\right)$$

## Results

### Invasion

For the mutant to invade, we must have $q'>q$ (the frequency of the mutation in the next generation must be higher than in the current generation). Solving for $q'>q$ in terms of $t$, we find:

$$t_{critical} > \frac{n}{2(n-1)} \quad \text{(S.3)}$$

Surprisingly, the critical value of $t$ allowing for invasion is independent of the degree of segregation infidelity ($f$) as long as f > 0 (*Figure 7B*).

In an attempt to get a more intuitive feel for the path a segregation infidelity mutant might take, we plotted trajectories of such mutants using variable parameters. *Figure 7C* shows that changes in some parameters influence the trajectories in predictable ways: a) increasing segregation infidelity from $f = 0.1$ to $f = 0.9$ allows for a more rapid spread; b) dominant mutations ($h = 1$) spread more rapidly than recessive ones ($h = 0$); c) more segregating drivers ($n = 5$) lead to a more rapid spread than fewer segregating drivers ($n = 3$); and d) lower strength of drive ($t = 0.55$ compared to $t = 0.95$) can lead to the loss of the infidelity mutant as predicted by *Equation S.3* and *Figure 7B*. Note that in this no-cost model, there is no stable internal equilibrium. All infidelity mutants are either lost or fixed. Accounting for additional costs of the infidelity mutant leads to the possibility of a stable equilibrium, though the parameters allowing this equilibrium are narrow (*Appendix 1—figure 1C*).

### Allowing for additional costs of segregation infidelity mutants

Segregation infidelity mutants incur an intrinsic fitness cost since 50% of the spores do not contain a chromosome. However, our model allows us to assess additional costs that this mutant may carry. We imagine two such costs. First, we allow for additional fitness costs ($s_s$) experienced by disomic spores. There is ample evidence that spores disomic for the third chromosome jettison one copy relatively quickly (*Niwa et al., 2006*), but there may be some initial cost due to gene dosage imbalance or other factors. Second, we can imagine that the meiotic mutant might also bear a cost because it leads to other problems including missegregation of the other chromosomes. We denote this cost as $s_m$ with an associated dominance ($h$). The dominance coefficient ($h$) is assumed to apply to both the cost of the segregation mutant ($s_m$) and the segregation infidelity ($f$). The recursion for the frequency of the infidelity mutant is therefore:

$$q' = \tfrac{1}{w}\left(2q(1-q)(1-hs_m)\left(\tfrac{1}{n}\left(\tfrac{(1-hf)}{2} + (1-s_s)\tfrac{hf}{4}\right)\right.\right.$$
$$\left. + (1-1/n)\left((1-t)\left(\tfrac{(1-hf)}{2} + (1-s_s)\tfrac{hf}{4}\right) + t(1-s_s)\tfrac{hf}{4}\right)\right)$$
$$+ q^2(1-s_m)\left(\tfrac{1}{n}\left((1-f) + (1+s_s)\tfrac{f}{2}\right)\right. \quad \text{(S.4A)}$$
$$\left.\left. + \left(1-\tfrac{1}{n}\right)\left((1-t)\left((1-f) + (1+s_s)\tfrac{f}{2}\right) + t\left((1+s_s)\tfrac{f}{2}\right)\right)\right)\right)$$

$$\bar{w} = (1-q)^2\left(\tfrac{1}{n} + \left(1-\tfrac{1}{n}\right)(1-t)\right) + 2q(1-q)(1-hs_m)$$
$$\left(\tfrac{1}{n}\left(1-hf+(1-s_s)\tfrac{hf}{2}\right) + (1-1/n)\left((1-t)\left(1-hf+(1-s_s)\tfrac{hf}{2}\right) + t(1-s_s)\tfrac{hf}{2}\right)\right) + q^2(1-s_m) \quad \text{(S.4B)}$$
$$\left(\tfrac{1}{n}\left((1-f)+(1+s_s)\tfrac{f}{2}\right) + \left(1-\tfrac{1}{n}\right)\left((1-t)\left((1-f)+(1+s_s)\tfrac{f}{2}\right) + t\left((1+s_s)\tfrac{f}{2}\right)\right)\right)$$

### Invasion with additional costs

For the mutant to invade, we again must have $q'>q$. Solving for $q'>q$ in terms of $t$, we find:

$$t_{critical} > \frac{n(2s_m + f(1-hs_m)(1+s_s))}{2(1-n)(s_m + f(1-hs_m))} \tag{S.5}$$

If we assume that there is no cost of being a disomic spore ($s_s = 0$), this becomes:

$$t_{critical} > \frac{-n(2s_m + f(1-hs_m))}{2(1-n)(s_m + f(1-hs_m))} \tag{S.5A}$$

If we instead assume that there is no cost of carrying the meiotic mutation ($s_m = 0$), this becomes:

$$t_{critical} > \frac{n(1+s_s)}{2(n-1)} \tag{S.5B}$$

*Appendix 1—figure 1B* shows the effect of these costs on the critical value of drive strength required for a meiotic infidelity mutant to invade. Note that even with high cost (20%), strong drivers select for these meiotic infidelity mutants. In addition to reducing the parameter space allowing for invasion of segregation infidelity mutants, these additional costs also slow the invasion of the mutant. For example, a mutant without extrinsic costs ($s_m$ and $s_s$ are 0) takes 25 generations to climb from a frequency of 1% to 50% ($f = 0.33$, $t = 0.9$, $h = 1$; $n = 5$). With a low cost ($s_m = 0.05$) this would take 32 generations, and with a higher cost ($s_m = 0.20$) this would take 755 generations. Note that with higher costs (e.g. $s_m = 0.3$), the mutant does not invade because our $t = 0.9$ is less than the critical $t = 0.978$ required for invasion. However, with observed parameters in various isolates of *S. pombe* ($t = 0.9$ and 4–14 distinct *wtf* drivers), the system could tolerate considerable cost and still allow mutants to invade.

## Equilibrium frequency of meiotic mutants with costs

Extrinsic costs of segregation infidelity mutants also allow for a narrow range of stable equilibria in some cases. Solving $q' = q$ for $q$ provides the equilibrium values. There are three such values: 0, 1 and

$$\hat{q} = \frac{h(2ns_m(t-1) - 2ts_m - f(q-hs_m)(n(1+s_s-2t)+2t))}{2s_m(2h-1)(nt-n-t) + f(1-s_m-2h(1-hs_m))(n(1+s_s-2t)+2t)} \tag{S.6}$$

The value obtained in *Equation S.6* is between zero and one for only a narrow range of parameter values. We can think of this in terms of a range of $t$ allowing for a stable equilibrium. Below $t_{critical}$, the mutation is lost. However, if $t$ is high enough, the meiotic mutation will fix. This happens when $q = 1$, which occurs when

$$t_{fixation} = \frac{n(-2s_m + f(s_m + hs_m - 1)(1+s_s))}{2(n-1)(-s_m + f(s_m + hs_m - 1))} \tag{S.7}$$

*Appendix 1—figure 1C* shows the narrow range allowing for stable equilibrium. Note that in the narrow space between the solid line for $t_{critical}$ and the dashed line for $t_{fixation}$, the stable equilibrium frequency ranges from zero to one.

## Limitations and caveats

Our model is simplistic in several ways but does suggest that a *wtf*-like poison-antidote system might select for mutations that disrupt 'normal' chromosome segregation fidelity. The first caveat is that we assume the population is randomly mating. However, due to spatial proximity of cells, we expect that inbreeding may be common. Such inbreeding would reduce the proportion of matings between haploids with different drive chromosomes and therefore decrease the benefit of segregation infidelity. A second caveat is that we assume all drive strengths are equal and all drivers are at equal frequency in the population. This is a logical starting point but is unlikely to be the case in real populations. Increasing the variance of the frequencies of individual drive chromosomes within a single population will also reduce the proportion of matings between haploids with different drive chromosomes and will decrease the benefit of segregation infidelity. Finally, we do not allow for non-driving third chromosomes. Diploids with a wild-type and a driving chromosome would still benefit from segregation infidelity. This benefit would be due to the disomic spores generated, in which

both poison and antidote would be produced, protecting the disomic spores from being killed. However, selection for the infidelity mutation would be weaker because the drive load in the population would be less since fewer spores are killed each generation due to drive.

