## [Decision Letter]

**Acceptance summary:**

This work demonstrates that parasitic genetic elements have the potential to affect the process of meiosis in a species. The authors show that prevalent aneuploidy in *S. pombe* allows spore survival in the presence of heterozygous meiotic drivers. The implication of this work is that imperfect chromosome segregation and meiotic drivers are co-dependent in evolution.

**Decision letter after peer review:**

Thank you for submitting your article "Atypical meiosis can be adaptive in outcrossed *S. pombe* due to *wtf* meiotic drivers" for consideration by *eLife*. Your article has been reviewed by three peer reviewers, including Adèle L Marston as the Reviewing Editor and Reviewer #1, and the evaluation has been overseen by Patricia Wittkopp as the Senior Editor. The following individual involved in review of your submission has agreed to reveal their identity: Thomas Price (Reviewer #2).

The reviewers have discussed the reviews with one another and the Reviewing Editor has drafted this decision to help you prepare a revised submission.

Summary:

Bravo Núñez et al. report that the meiotic drivers of the *wtf* family in *S. pombe* benefit from impaired meiotic chromosome segregation to be maintained in the population. This work extends the seminal work from Zanders et al., 2014, who first reported the importance of meiotic drive as a source of hybrid sterility in *S. pombe*, and the prevalence of diploid and aneuploid spores in such hybrids. Here, the authors found that any mean to maintain heterozygosity of the *wtf* alleles increases spore viability and hence can be selected. This applies to meiotic mutants such as *rec8*, the meiotic specific kleisin subunit of the cohesion complex, but also to chromosome configurations prone to unequal crossovers and hence gene duplications. The authors found that the frequency of aneuploid spores in homozygous *S. pombe* isolates is about 5% or higher, which is a situation rather permissive to accumulate aneuploid spores maintaining *wtf* heterozygosity. These findings are interesting, but the presented interpretation is open to debate, and other possibilities need to be considered. It is also unclear whether the assays used are reporting on meiotic drive or instead some alternation in the process of meiosis itself.

Essential revisions:

1) The authors present the high aneuploidy as a consequence of the presence of the *wtf* element. However, from the presented data, it is not possible to distinguish between a consequence of the presence of the *wtf* elements or a cause for the presence and maintenance of these elements in *S. pombe*. Similarly, the title implies *wtf* drivers are the cause of adaptive/abnormal meiosis while it could very well be the other way around: *wtf* drivers may have spread in *S. pombe* because of inherently high level of aneuploidy. Stronger evidence is needed to say the high missegregation rates in *S. pombe* have actually evolved in response to *wtf* elements. Both possibilities are interesting but the authors need to present clear evidence to distinguish between them.

2) The paper needs major re-writing to gain in clarity and transparency. The rationale is sometimes hard to follow and the manuscript is difficult to read, even for specialists. For instance, the last figure (Figure 8) could very well be the first figure of the paper because it illustrates the high frequency of aneuploid spores in *all* homozygous *S. pombe* isolates tested, which seems a pre-requisite for the maintenance of the *wtf* elements. Also, assays are presented in a very technical way. Specific examples are given below, but this consideration should be applied to the entire manuscript including parameters used as measurements in assays need to be defined upfront.

3) Some of the observations in the current manuscript were already published in Zanders et al., 2014, and in a much clearer way. The novelty with respect to Zanders et al., 2014 needs to be made precisely clear at the outset.

4) If chromosome segregation errors offer protection against *wtf* drivers, why is this phenomenon not observed more generally with all the chromosome segregation mutants tested. Notably, it is concerning that *rec12* and *rec8* were particularly protective as both of these mutations affect meiotic recombination.

5) The conclusions drawn in this manuscript rely on there being no increase in the number of disomic spores generated in total, rather than just an increase in the number of disomic spores that survive. It is possible that the *wtf* genes disturb some aspect of meiosis (in particular meiotic recombination as they will increase heterozygosity). However, this was not considered in the present manuscript and for the set of strains generated here. If heterozygous *wtf* drivers decrease the number of crossovers, then homologs would mis-segregate in the first division resulting in an increase in heterozygous disomic spores. Can this be ruled out?

6) The authors conclusions are based on back-projections/inference from random spore survival and therefore the measurements are rather indirect. Can the authors be sure that the observed effects are due to spore killing rather than some other phenomenon that occurs during the meiotic programme. Better analysis would be to perform tetrad analysis where chromosome segregation patterns and spore killing can be followed for the same meiotic event.

7) One would also expect a better discussion about the implications of the current findings at the population level:

i) The high aneuploidy rate in homozygous backgrounds, as well as the selection for disomy for chromosome 3 in hybrids, suggests that aneuploidy for chromosome 3 should be frequent at the population level. What's the situation exactly?

ii) It seems that meiotic recombination is playing only a minor role in spore viability in hybrids and a limited role in homozygous isolates: what is the selective pressure on meiotic recombination in *S. pombe*? Is there any? Is meiotic recombination about to be lost? Is outcrossing so infrequent that the only selective pressure comes from homozygous isolates?

iii) The strength of the *wtf* drivers is so strong that one would expect it to invade the whole population rather quickly. If this is not the case, does it argue for infrequent (absent?) outcrossing in natural populations?

iv) Can the high aneuploidy rate be related to specific variants of meiotic genes, notably for strain JB848?

---

## [Author Response]

Essential revisions:1) The authors present the high aneuploidy as a consequence of the presence of the wtf element. However, from the presented data, it is not possible to distinguish between a consequence of the presence of the wtf elements or a cause for the presence and maintenance of these elements in *S. pombe*. Similarly, the title implies wtf drivers are the cause of adaptive/abnormal meiosis while it could very well be the other way around: wtf drivers may have spread in *S. pombe* because of inherently high level of aneuploidy. Stronger evidence is needed to say the high missegregation rates in *S. pombe* have actually evolved in response to wtf elements. Both possibilities are interesting but the authors need to present clear evidence to distinguish between them.

We demonstrate that heterozygous *wtf* drivers select for disomic spores and spores with *wtf* duplications. We also demonstrate with empirical and population genetic modeling that competing *wtf* genes can make mutations that decrease meiotic fidelity selectively advantageous. Our data thus fully support our title: Atypical meiosis can be adaptive in outcrossed *S. pombe* due to *wtf* meiotic drivers.

We also show that *S. pombe* makes disomic spores at a relatively high frequency. It is possible that a history of *wtf* competition drove the evolution of low meiotic fidelity in *S. pombe* and we present this as a possibility. The reviewers are correct that we have not proven that model and we have added to the Discussion (subsection “Meiotic drivers shape the evolution of *S. pombe* meiosis”, last paragraph) to clarify that we do not claim alternative models are not possible. This section includes discussion of the alternative model that *wtf* drivers are successful in *S. pombe* because of high levels of aneuploid gametes. Finally, this section provides more information about how little is known about *S. pombe* ecology and evolution to clarify why it is not currently possible to distinguish between the models.2) The paper needs major re-writing to gain in clarity and transparency. The rationale is sometimes hard to follow and the manuscript is difficult to read, even for specialists. For instance, the last figure (Figure 8) could very well be the first figure of the paper because it illustrates the high frequency of aneuploid spores in ALL homozygous *S. pombe* isolates tested, which seems a pre-requisite for the maintenance of the wtf elements. Also, assays are presented in a very technical way. Specific examples are given below, but this consideration should be applied to the entire manuscript including parameters used as measurements in assays need to be defined upfront.

We have extensively rewritten the paper, especially the Results, to better explain the assays and to improve clarity. As suggested, we now present the data from what was Figure 8 earlier in the paper. It is now Figure 1—figure supplement 3. It is not, however, clear that a high frequency of aneuploid spores is a pre-requisite for maintenance of *wtf* elements. Rather, aneuploidy provides a mechanism for suppressing drive (reduces allele transmission bias). This suppression can benefit drivers only in the sense that it prevents them from destroying themselves by causing complete infertility. We have tried to clarify this point as well in the Discussion (subsection “Meiotic drivers shape the evolution of *S. pombe* meiosis”, last paragraph).

3) Some of the observations in the current manuscript were already published in Zanders et al., 2014, and in a much clearer way. The novelty with respect to Zanders et al., 2014 needs to be made precisely clear at the outset.

We tried to clarify this in the text (see the first section of the Results). In Zanders et al., 2014, we investigated *Sk*/*Sp*, *Sk*/*Sk* and *Sp*/*Sp* diploids in *rec12*+ and *rec12*- backgrounds. We observed high aneuploidy, low fertility and drive in the hybrids. We also found *rec12*+ did not promote fertility in the hybrids. We cite the previous paper and mention those observations were first described in Zanders et al., 2014. In that work, however, we did not know what genes caused drive or how complex their landscape was. We also did not prove the connection between aneuploidy and drive, although we speculated the connection existed.

The starting point of this work was to understand the phenotypes of the *Sk*/*Sp* heterozygotes. Because of this, it was essential that we discuss those phenotypes. We also wanted to be sure that the phenotypes were generalizable to the *S. pombe* species, so we repeated similar analyses in other heterozygotes. We view this as adding rigor to the work, rather than undermining the novelty.

4) If chromosome segregation errors offer protection against wtf drivers, why is this phenomenon not observed more generally with all the chromosome segregation mutants tested. Notably, it is concerning that rec12 and rec8 were particularly protective as both of these mutations affect meiotic recombination.

We chose not to discuss the meiotic mutants in depth to focus on conveying the main idea that the fitness costs of meiotic mutants can be different when *wtf* genes compete (e.g. inbreeding vs. outcrossing). This point is well supported by our data.

We agree that understanding the precise differences between the various mutant phenotypes would be very interesting. However, this would require more extensive analyses beyond the scope of this work. This is because in the *wtf4*/*wtf28* heterozygotes (Figure 6), two types of events can lead a spore to inherit both *wtf4* and *wtf28*: an uneven crossover (CO) and chromosome nondisjunction. Our data in Figure 4 shows that more than half the viable spores produced by *wtf4/wtf28* heterozygotes result from uneven COs. In *wtf4/wtf28* heterozygotes, *rec12* actually promotes fertility (Figure 6—figure supplement 1). We do not know how the other mutants tested affect the frequency of the uneven CO event and assaying for the uneven CO in additional mutant backgrounds would not be trivial.

If we wanted to discuss the mutants in more depth, we would also need to also analyze the mutants presented in Figure 6 in diploids generated by outcrossing or in diploids with *wtf* drivers competing at two loci, as in Figure 2. In those diploids, COs are very unlikely to generate a spore that inherits all *wtf* drivers and *rec12* does not promote fertility (Figure 2, compare diploid 13 to diploid 15).

5) The conclusions drawn in this manuscript rely on there being no increase in the number of disomic spores generated in total, rather than just an increase in the number of disomic spores that survive. It is possible that the wtf genes disturb some aspect of meiosis (in particular meiotic recombination as they will increase heterozygosity). However, this was not considered in the present manuscript and for the set of strains generated here. If heterozygous wtf drivers decrease the number of crossovers, then homologs would mis-segregate in the first division resulting in an increase in heterozygous disomic spores. Can this be ruled out?

Our data support that there is no increase in disomic spores generated per meiosis when *wtf* drivers are heterozygous (Figure 2—figure supplement 4), compared to when they are homozygous. This suggests the increased disomic fraction in *wtf* heterozygotes is due to the death of haploid spores. We have tried to clarify these data and the implications in the text (subsection “Multiple sets of competing meiotic drivers can select for disomic spores by killing haploids”, fourth paragraph). In addition, we have provided a tetrad-based assay of spore death as a supplementary figure (Figure 4—figure supplement 1). If spore death in the *wtf4*/*wtf28* heterozygotes was due to chromosome nondisjunction, we would anticipate the classic 4:2:0 pattern of spore death (i.e. tetrads with 4, 2, and 0 dead spores). Instead, we observe mostly tetrads with 4 dead spores and few 2-spore viable tetrads (Figure 4—figure supplement 1).

We also showed that *wtf* competition contributes to high disomy in both *rec12*+ and *rec12*- backgrounds (see Figure 2 and Figure 3). This shows that *wtf* driver competition contributes to disomy independent of potential changes in recombination.

It is quite possible that if *wtf* drivers were frequently heterozygous, that could put selective pressure on meiosis to change recombination frequencies. Such changes, however, would likely be due to selection for variants in the recombination machinery/chromosome structural variants, rather than any direct involvement of Wtf proteins in recombination. Low levels of cytoplasmic Wtf^poison^ protein can be detected in cells prior to meiosis I, so the proteins could theoretically affect chromosome segregation (Nuckolls et al., 2017). However, a highly variable number of *wtf* drivers are always present in *S. pombe* meiosis. It is not clear how the cell would be able to distinguish *wtf* heterozygosity from variation in *wtf* gene dosage during recombination. It is also likely *wtf* gene dosage does not affect recombination because measured recombination frequencies are similar between the lab isolate and *Sk*, even though *Sk* has 10 drivers and the lab isolate has 4 (Zanders et al., 2014, Eickbush et al., 2019). Finally, we added additional analyses showing that Wtf poison proteins, which are present during meiosis, do not affect disomy frequencies independent of drive (subsection “Multiple sets of competing meiotic drivers can select for disomic spores by killing haploids”, third paragraph; Figure 2—figure supplement 3). We feel that our interpretation, which requires only spore killing, a known phenotype of *wtf* drivers, is the most parsimonious and is well supported by the data.

6) The authors conclusions are based on back-projections/inference from random spore survival and therefore the measurements are rather indirect. Can the authors be sure that the observed effects are due to spore killing rather than some other phenomenon that occurs during the meiotic programme. Better analysis would be to perform tetrad analysis where chromosome segregation patterns and spore killing can be followed for the same meiotic event.

As discussed above, we now include a tetrad-based example of *wtf* driver-induced spore death (Figure 4—figure supplement 1) that supports our interpretation.

Tetrad dissections, however, are problematic for studying *wtf* drive because there are almost never 4 spores that can be dissected. Drive does not just kill spores, it destroys them. Picking asci with 4 things that might have been spores is possible when one driver is acting, but this biases analyses to asci produced by meioses with more mild drive phenotypes. When more than one driver is acting, picking spores is even harder and they often fall apart when one gets them on the needle. The sorry state of the spores destroyed by *wtf* drivers is apparent in Figure 4—figure supplement 1B-D.

7) One would also expect a better discussion about the implications of the current findings at the population level:i) The high aneuploidy rate in homozygous backgrounds, as well as the selection for disomy for chromosome 3 in hybrids, suggests that aneuploidy for chromosome 3 should be frequent at the population level. What's the situation exactly?

In all cases, the aneuploids we detect are unstable. By the time the spore grows into a colony, most of the cells in the colony are haploid. We can tell that the spore that started the colony was disomic based on the presence of the two centromere-linked markers. Similarly, almost all sampled isolates of *S. pombe* are haploid. We have clarified that the aneuploids are unstable (subsection “Equilibrium frequency of meiotic mutants with costs”).

ii) It seems that meiotic recombination is playing only a minor role in spore viability in hybrids and a limited role in homozygous isolates: what is the selective pressure on meiotic recombination in *S. pombe*? Is there any? Is meiotic recombination about to be lost? Is outcrossing so infrequent that the only selective pressure comes from homozygous isolates?

In homozygous isolates, *rec12* deletion reduces spore viability to about 1/3 of the wild-type level. Even ignoring Muller’s ratchet, this would suggest in homozygotes there is a strong selection to maintain recombination. In hybrids, it appears dispensable. To understand what the overall selective pressures might be, we would need more ecological data about population structure and outcrossing rates. We now explain these challenges in the Discussion (subsection “Meiotic drivers shape the evolution of *S. pombe* meiosis”, last paragraph). Molecular evolution analyses comparing meiotic genes to non-meiotic genes could also shed light on this question, but those analyses are beyond the scope of this work.

iii) The strength of the wtf drivers is so strong that one would expect it to invade the whole population rather quickly. If this is not the case, does it argue for infrequent (absent?) outcrossing in natural populations?

This is an interesting question, but the answer is complex and beyond the scope of this work. In short, the *wtf* genes do not follow the assumed rules governing meiotic drivers, so it is hard to use those rules to guess at outcrossing frequencies. Ecological information is required, as we now mention in the Discussion (subsection “Meiotic drivers shape the evolution of *S. pombe* meiosis”, last paragraph). However, all sequenced isolates of *S. pombe* do contain *wtf* genes. *wtf* genes have only been assembled in a limited number of isolates, but each of those isolates contains at least four intact *wtf* drivers, in addition to many more *wtf* genes that cannot drive autonomously. All of the drivers are very strong, but the sequences of the drivers are generally different between strains, as are the loci where the drivers are found. Extremely rapid evolution (largely due to nonallelic gene conversion within the family) leads to constant birth and death of *wtf* drivers (Eickbush et al., 2019, Bravo Núñez et al., 2020). It is like driver whack-a-mole in the *S. pombe* genome.

iv) Can the high aneuploidy rate be related to specific variants of meiotic genes, notably for strain JB848?

Identifying the variants underlying the differences in disomic gamete frequency is beyond the scope of this work. We are, however, currently investigating this question and have not found any obvious causes.